# CoT Vectors: Transferring and Probing the Reasoning Mechanisms of LLMs

**Li Li**[1]  **Ziyi Wang**[1]  **Yongliang Wu**[1]  **Jianfei Cai**[2]  **Xu Yang**[1]*
[1] School of Computer Science & Engineering, Key Laboratory of New Generation Artificial Intelligence Technology and Its Interdisciplinary Applications (Southeast University), Ministry of Education, China
[2] Data Science & AI Department at Faculty of IT, Monash University, Australia
`lilyli@seu.edu.cn, xuyang_palm@seu.edu.cn`

## Abstract

Chain-of-Thought (CoT) prompting has emerged as a powerful approach to enhancing the reasoning capabilities of Large Language Models (LLMs). However, existing implementations, such as in-context learning and fine-tuning, remain costly and inefficient. To improve CoT reasoning at a lower cost, and inspired by the task vector paradigm, we introduce CoT Vectors, compact representations that encode task-general, multi-step reasoning knowledge. Through experiments with Extracted CoT Vectors, we observe pronounced layer-wise instability, manifesting as a U-shaped performance curve that reflects a systematic three-stage reasoning process in LLMs. To address this limitation, we propose Learnable CoT Vectors, optimized under a teacher–student framework to provide more stable and robust guidance. Extensive evaluations across diverse benchmarks and models demonstrate that CoT Vectors not only outperform existing baselines but also achieve performance comparable to parameter-efficient fine-tuning methods, while requiring fewer trainable parameters. Moreover, by treating CoT Vectors as a probe, we uncover how their effectiveness varies due to latent space structure, information density, acquisition mechanisms, and pre-training differences, offering new insights into the functional organization of multi-step reasoning in LLMs. The source code will be released.

## 1 Introduction

Chain-of-Thought (CoT) prompting (Wei et al., 2022) has emerged as a powerful technique to unlock the complex reasoning capabilities of Large Language Models (LLMs) (Zhao et al., 2023). By reasoning step-by-step, CoT enables models to decompose problems, mimic human-like logic, and improve performance on several challenging tasks (Imani et al., 2023; Huang & Chang, 2022). However, how to effectively harness the power of CoT in practice remains an open problem. Existing approaches generally fall into two categories: (1) In-Context Learning (ICL) (Brown et al., 2020) with few-shot CoT examples, which can enhance reasoning, but it requires longer prompts and slows inference; (2) Fine-tuning LLMs (Ziegler et al., 2019) with CoT-annotated data, which demands large amounts of high-quality reasoning traces and computational resources, while often yielding only limited improvements for models that already equipped with CoT abilities. These challenges prompt a critical question: *can we transfer the essence of CoT, i.e., the general "problem-solving mindset" of a task, into LLMs in a way that is compact, reusable, and efficient?*

Recent advances in Task Vectors (Ilharco et al., 2022) offer a promising direction. Task-specific knowledge can be distilled into a compact vector, often represented as the difference in activations (Hendel et al., 2023; Todd et al., 2023; Liu et al., 2023) or parameters (Ortiz-Jimenez et al., 2023b; Li et al., 2025) between fine-tuned and base models. Such vectors can steer model behavior toward new tasks without modifying model weights, thereby enabling parameter-efficient adaptation. However, current applications of Task Vectors have been limited to simple adaptation scenarios, leaving it unclear whether this paradigm can be extended to complex multi-step reasoning.

---

*Corresponding author.

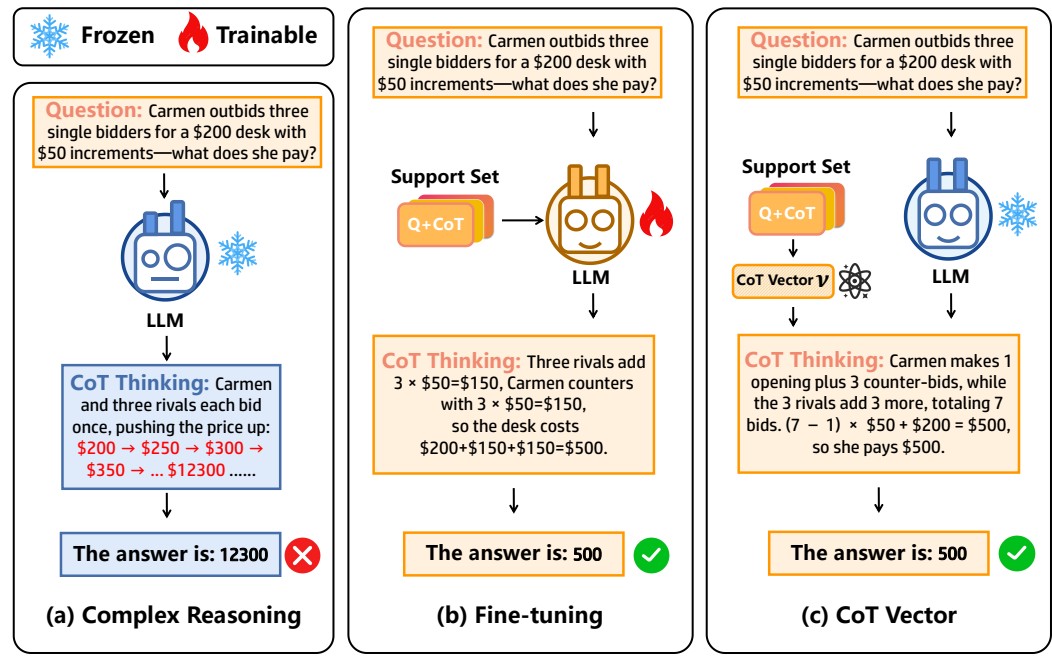

Figure 1: Overview of our approach. (a) Standard LLM may struggle to produce a correct reasoning chain for a complex problem. (b) Conventional fine-tuning adapt the model to such tasks by training on a support set, but requires updating model parameters, incurring high computational cost. (c) Our proposed CoT Vector leverages the support set to obtain a compact reasoning representation, which can be injected into the forward process of model to guide reasoning efficiently.

Through mathematical analysis, we observe that the effect of CoT can be formalized as a consistent shift in the internal activations of model (He et al., 2021), suggesting that extending task vectors to reasoning is both feasible and promising. In this work, we introduce CoT Vectors, task-general reasoning representations that adapt the task vector framework to CoT reasoning. CoT Vectors compactly encode the critical reasoning knowledge from a support set of (Question, CoT, Answer) triplets, and can be directly injected into the forward process during inference. This approach not only enables portable reasoning enhancement without costly retraining or significant inference overhead, but also offers a new probe into how LLMs internalize and apply CoT. Specifically, we begin with Extracted CoT Vectors, directly derived from activation differences between reasoning and non-reasoning traces, in line with the traditional task vector approach in NLP. Our studies reveal that Extracted CoT Vectors are effective but highly unstable across layers, with a striking U-shaped performance curve. This pattern suggests a systematic functional organization in LLMs, which we characterize as a three-stage reasoning process spanning perception, reasoning, and expression. Shallow and deep layers show relatively consistent representations, whereas middle layers contain highly variable, sample-specific structures that cause extracted vectors to fail.

To improve robustness, we introduce Learnable CoT Vectors, optimized via a teacher–student framework. Mathematically inspired by the additive shift formalization of CoT, our method distills a more robust and generalizable reasoning signal into a single, reusable vector. By actively learning reasoning knowledge rather than passively averaging activations, it achieves greater stability and stronger performance, overcoming the layer-wise volatility of extracted vectors. We conduct a comprehensive evaluation across various models and benchmarks, comparing extracted and learnable vectors against baselines. Our analyses further elucidate the sources of variability in CoT Vector effectiveness, highlighting how differences in acquisition mechanisms and model-specific latent structures shaped during pre-training impact reasoning performance. This perspective offers a valuable lens for understanding how LLMs organize and apply multi-step reasoning internally.

Overall, we summarize the main contributions of this work as follows:

- We introduce CoT Vectors, extending task vectors to multi-step reasoning. Experiments with traditional extracted vectors uncover their layer-wise instability, which in turn reveals a systematic three-stage reasoning process in LLMs.

- To address the limitations of extracted vectors, we propose novel Learnable CoT Vectors, optimized via a teacher–student framework, which provide more robust, stable, and task-general reasoning representations.
- We conduct a comprehensive evaluation across benchmarks and models. Using CoT Vectors as a probe, we analyze their variability from multiple perspectives, including latent space structure, information density, acquisition mechanisms, and model pre-training differences, thereby providing new insights into the mechanistic organization of reasoning in LLMs.

## 2 RELATED WORK

**Enhancing Chain-of-Thought Reasoning in LLMs.** Our work builds upon the foundational paradigm of CoT prompting (Wei et al., 2022), which significantly improves reasoning performance by eliciting step-by-step rationales. Numerous subsequent efforts have refined this idea through improved prompting strategies (Kojima et al., 2022; Khot et al., 2022), search-based reasoning frameworks (Yao et al., 2023), program-aided execution (Gao et al., 2023), and iterative self-refinement (Madaan et al., 2023). Another common strategy to improve performance is fine-tuning. Typical approaches include Supervised Fine-tuning (SFT) on (Question, CoT, Answer) triplets, as well as reinforcement learning techniques such as RLHF (Ouyang et al., 2022), PPO (Schulman et al., 2017), and GRPO (Rafailov et al., 2023). While effective, these techniques often demand substantial amounts of high-quality rationales, significant computational resources for training or alignment (Zhao et al., 2025), making them prohibitively expensive relative to the incremental gains.

A parallel line of work seeks to compress explicit reasoning steps into fewer, or even invisible, latent representations, often referred to as Implicit CoT (Deng et al., 2023; 2024). These methods aim to internalize the reasoning process to improve efficiency and performance. Some approaches modify model architecture (Geiping et al., 2025) or use placeholder tokens in prompts (Pfau et al., 2024) to extend latent reasoning depth, effectively condensing multi-step thinking into compressed latent transitions that lead directly to answers. However, these methods typically require specialized model modifications and carefully engineered training regimes involving intensive post-training of model parameters (Hao et al., 2024; Shen et al., 2025; Cheng & Van Durme, 2024), which demand substantial resources while often delivering only modest gains. In contrast, our approach keeps the model architecture untouched: we distill reasoning into an external, plug-and-play CoT Vectors that can be acquired and applied rapidly for new tasks, combining flexibility with strong performance.

**Task vectors.** Task vectors (Ilharco et al., 2022) have emerged as a compact representation of task-specific knowledge, typically obtained either by computing weight differences between fine-tuned and pre-trained models (Ortiz-Jimenez et al., 2023a; Li et al., 2025), or by capturing activation differences induced by distinct input prompts (Liu et al., 2023; Todd et al., 2023; Hendel et al., 2023). These vectors not only enable parameter-efficient task transfer but have also been leveraged to provide preliminary insights into the internal mechanisms of LLMs (Yang et al., 2025). However, existing studies largely focus on relatively simple scenarios such as classifications or in-context learning, leaving the application of task vectors to complex multi-step reasoning underexplored. Several recent preliminary study (Azizi et al., 2025; Tang et al., 2025; Zhang & Viteri) have tentatively explored steering vectors in CoT, suggesting the feasibility of the paradigm. However, Azizi et al. (2025) focuses on compressing CoT chains, while Tang et al. (2025) aims to stimulate longer reasoning trajectories, both focusing on controlling CoT generation rather than capturing a task-general reasoning pattern. Meanwhile, Zhang & Viteri remains limited to conventional extraction techniques and offered only surface-level analysis. Our work moves substantially beyond this early exploration. Instead of relying solely on a basic extraction method, we introduce a novel learnable mechanism that actively optimizes CoT Vectors for better generalization and performance. Furthermore, our study provides a comprehensive analysis absent from prior work.

## 3 METHODOLOGY

We first formalize the concept of CoT Vectors in Section 3.1, deriving it from the mechanistic effect of CoT reasoning on the model's attention outputs. This formulation not only establishes the feasibility of our approach but also provides the guiding principle for the subsequent development. Building on this, we develop our two practical frameworks in Section 3.2 for acquiring CoT Vectors:

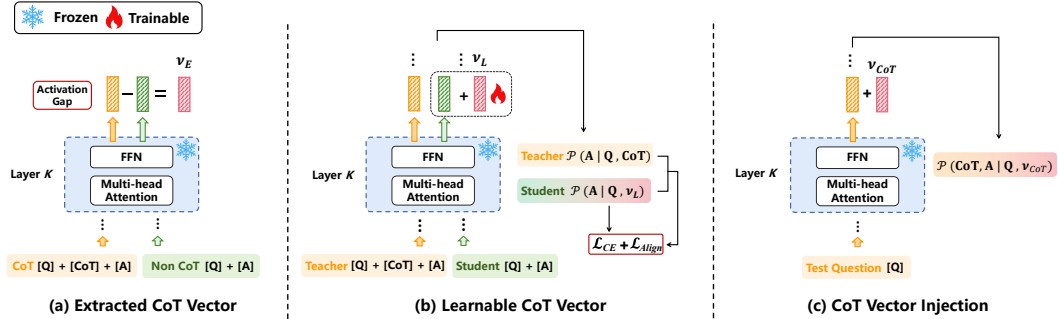

Figure 2: Methods for acquiring and applying CoT Vectors. (a) Extracted CoT Vector is obtained by recording the activation gap at the $k$-th layer between inputs with and without CoT. (b) Learnable CoT Vector is inserted into the $k$-th layer activations of a student sequence without CoT, and trained by aligning the student's final answer-token hidden states with those of a teacher sequence that includes CoT. (c) At test time, CoT Vector is added to the activations at the $k$-th layer during each forward pass of auto-regressive generation, guiding the reasoning process.

a non-parametric extraction method and a novel parametric learning-based method, along with how to efficiently integrate the resulting vectors during inference to steer the model's reasoning.

## 3.1 CONCEPTUALIZATION AND FORMALIZATION OF COT VECTOR

The effectiveness of CoT prompting highlights that inserting an explicit reasoning sequence between the input question $Q$ and the final answer $A$ significantly enhances the model's reasoning capability. Our objective is to capture and quantify the effect of this reasoning process within the model.

In transformer-based language models, information flow is governed by self-attention. He et al. (2021) suggest that the effect of input prefixes can be understood as shifts in the attention outputs. In the context of CoT, the CoT sequence can be viewed as a specialized prefix that modulates the generation of answer tokens. When reasoning with CoT, the generation of an answer token depends not only on the question tokens but also on the intermediate CoT tokens. For each answer token $a \in A$, the single-head self-attention with and without the CoT sequence is denoted as $\text{SA}(a, [K_Q, K_C, K_A], [V_Q, V_C, V_A])$ and $\text{SA}(a, [K_Q, K_A], [V_Q, V_A])$ respectively, where the subscripts $Q$, $C$, and $A$ refer to the question, CoT, and answer. We then derive the following equation:

$$\text{SA}(a, [\boldsymbol{K}_Q, \boldsymbol{K}_C, \boldsymbol{K}_A], [\boldsymbol{V}_Q, \boldsymbol{V}_C, \boldsymbol{V}_A]) = \underbrace{\text{SA}(a, [\boldsymbol{K}_Q, \boldsymbol{K}_A], [\boldsymbol{V}_Q, \boldsymbol{V}_A])}_{\text{Standard attention}} \tag{1}$$

$$+ \mu \cdot \underbrace{(\text{SA}(a, [\boldsymbol{K}_C], [\boldsymbol{V}_C]) - \text{SA}(a, [\boldsymbol{K}_Q, \boldsymbol{K}_A], [\boldsymbol{V}_Q, \boldsymbol{V}_A]))}_{\text{CoT shift}}$$

$$\tag{2}$$

The introduction of CoT induces an additional term in the attention output, whose influence is quantified by a scalar coefficient $\mu$ (see the supplementary material for the full derivation). This additional contribution reflects precisely the knowledge injected by the CoT sequence. We formalize this effect as a CoT Shift, and denote the corresponding representation as the CoT Vector $\vec{v}_{\text{CoT}}$. Accordingly, Equation 1 can be reformulated as

$$\text{SA}(a, [\boldsymbol{K}_Q, \boldsymbol{K}_C, \boldsymbol{K}_A], [\boldsymbol{V}_Q, \boldsymbol{V}_C, \boldsymbol{V}_A]) = \text{SA}(a, [\boldsymbol{K}_Q, \boldsymbol{K}_A], [\boldsymbol{V}_Q, \boldsymbol{V}_A]) + \mu \cdot \vec{v}_{\text{CoT}} \tag{3}$$

The CoT Vector serves as a compact representation of the reasoning knowledge compressed from the CoT sequence. We hypothesize that, for tasks of the same type, the CoT Vectors derived from individual examples reside in a continuous semantic space. The centroid of this space, which we call the task-general CoT Vector, encodes the shared reasoning strategy for that task. For a new problem, injecting $\vec{v}_{\text{CoT}}$ into the model's forward pass, which reversely applies the Equation 3, can effectively guide the model toward an appropriate reasoning trajectory and thereby improves task accuracy.

### 3.2 TASK-GENERAL CoT VECTORS

To leverage the advantages of task-general CoT Vectors, we first acquire them from a support set $D$. We propose two approaches for this acquisition: a traditional extraction-based method and a novel parametric learning-based method. Once obtained, the task-general vector is injected into the model's forward pass during inference to steer its reasoning process.

#### 3.2.1 EXTRACTED CoT VECTORS

Given a support set of pairs $(Q, A)$ and triplets $(Q, \text{CoT}, A)$, we define the Extracted CoT Vector $\vec{v}_{\text{CoT}}^{(l)}$ as the difference in model activations of answer token $a$ at layer $l$ between these inputs, as shown in Figure 2 (a):

$$\vec{v}_{\text{CoT}}^{(l)} = \frac{1}{|A|} \sum_{a \in A} \left( \boldsymbol{\alpha}_{\text{CoT}}^{(l)}(a) - \boldsymbol{\alpha}_{\text{Non-CoT}}^{(l)}(a) \right) \tag{4}$$

where $\boldsymbol{\alpha}^{(l)}(a)$ is the hidden state of answer token $a$ at layer $l$ for the input and $|A|$ denotes the total number of answer tokens.

For each instance $(Q_i, \text{CoT}_i, A_i)$, we compute an instance-specific CoT Vector $\vec{v}_{\text{CoT},i}$. A task-general Extracted CoT Vector $\vec{v}_E$ is then obtained by averaging across all $N$ support instances:

$$\vec{v}_E = \frac{1}{N} \sum_{i=1}^{N} \vec{v}_{\text{CoT},i} \tag{5}$$

#### 3.2.2 LEARNABLE CoT VECTORS

Beyond extraction, we propose a novel parametric method that learns a task-general CoT Vector through gradient-based optimization. As depicted in Figure 2 (b), $\vec{v}_L$ is initialized as learnable parameters, added as a shift to the hidden state at a specific layer, and optimized on the support set $\mathcal{D}$ to encode generalized reasoning knowledge. We adopt a teacher–student framework. For each instance $(Q_i, \text{CoT}_i, A_i)$, the teacher path processes the full triplet with frozen model parameters, providing the supervisory signal. The student path, in contrast, only processes $(Q_i, A_i)$, while $\vec{v}_L$ is injected to compensate for the missing CoT sequence. Through this process, $\vec{v}_L$ distills essential reasoning signals from the teacher into a compact, transferable representation.

Throughout optimization, all original LLM parameters are kept frozen; only $\vec{v}_L$ are updated. The training objective combines two components: Prediction loss ($\mathcal{L}_{\text{CE}}$) is the cross-entropy loss on the student's predicted answer tokens, ensuring that the injected vector guides the model toward correct outputs. Representation alignment loss ($\mathcal{L}_{\text{Align}}$) is the mean KL loss between hidden states of teacher and student paths at the answer tokens, enforcing alignment of internal reasoning representations. The final objective is:

$$\mathcal{L} = \mathcal{L}_{\text{Align}} + \lambda \cdot \mathcal{L}_{\text{CE}} \tag{6}$$

where $\lambda$ is a hyperparameter balancing the two terms.

#### 3.2.3 INTEGRATING THE CoT VECTOR TO REASONING

At inference time, given a new question, task-general CoT Vector $\vec{v}_{\text{CoT}}$ obtained from the support set at specific layer $l$, is then injected into the model at the same layer during every forward pass of CoT thinking, as shown in Figure 2 (c).

$$\tilde{\boldsymbol{\alpha}}^{(l)} = \boldsymbol{\alpha}^{(l)} + \mu^{(l)} \cdot \vec{v}_{\text{CoT}}^{(l)} \tag{7}$$

For Extracted CoT vectors, $\mu$ is an explicitly defined constant scaling factor. For Learnable CoT Vectors, however, $\mu$ is effectively internalized—since $\vec{v}_L$ is optimized end-to-end, the scaling factor is absorbed into the vector during training rather than being maintained as a separate constant. This integration incurs almost no additional overhead: it does not increase the input context length, and the runtime cost is negligible since the operation reduces to a simple vector addition. As a result, our approach provides an extremely efficient mechanism for enhancing reasoning in LLMs.

# 4 EXPERIMENTS

In this section, we first outline the setup and implementation details (Section 4.1). Next, we explore the adaptation of task vectors to multi-step reasoning in LLMs by introducing and analyzing CoT Vectors (Section 4.2). This investigation extends beyond mere performance evaluation, utilizing CoT Vectors as a tool to probe the underlying functional mechanisms of reasoning within LLMs.

## 4.1 SETUP AND IMPLEMENTATION DETAILS

**Models and Datasets.** We conduct experiments on two open-source LLMs: Qwen2.5-Math-7B (Yang et al., 2024), a model fine-tuned for mathematical reasoning tasks, and LLaMA-3.1-8B-Instruct (Grattafiori et al., 2024), an instruction-tuned model with broad domain coverage. We use three datasets with ground-truth CoT sequences: GSM8K (Cobbe et al., 2021), MATH (Hendrycks et al., 2024), MMLU-Pro (Wang et al., 2024), CommonsenseQA (Talmor et al., 2019), and StrategyQA (Geva et al., 2021). GSM8K and MATH cover mathematical reasoning, MMLU-Pro spans diverse subject domains, and CommonsenseQA/StrategyQA focus on natural-language logical reasoning. For MATH, we divide the dataset into two subsets based on difficulty: MATH-Easy (levels 1–3) and MATH-Hard (levels 4–5), ensuring balanced difficulty in the support set. We sample 3,000 examples from GSM8K and the two MATH subsets as the support set. For MMLU-Pro, we use all 70 ground-truth annotated problems as the support set. For CommonsenseQA and StrategyQA, no official CoT traces are provided, so we rely on model-generated CoT sequences instead.

**Implementation Details.** We use standard zero-shot CoT prompting as baseline, where the model is instructed to "think step by step." For CoT Vectors, both extracted and learnable variants are implemented following the procedures described in Section 3. For extracted vectors, the scaling factor $\mu$ is fixed at 1.0. For learnable vectors, the loss balancing factor $\lambda$ in Eq. 6 is set to 0.5. We select LoRA (Hu et al., 2022) as the representative parameter-efficient fine-tuning baseline, where LoRA adapters are trained on $(Q, \text{CoT}, A)$ triplets. Following common practice, we apply LoRA to the projection matrices $W_Q, W_K, W_V$, and $W_O$ in all attention layers.

## 4.2 EXPLORING CoT VECTORS AND THE MECHANISM OF REASONING

This section explores the adaptation of task vectors to multi-step reasoning via CoT Vectors. Our analysis begins with examining CoT Vectors obtained by the traditional extraction method, which prove effective but highly unstable performance across layers. This instability reveals consistent layer-wise patterns, uncovering a three-stage reasoning process in LLMs. Building upon these insights, we introduce Learnable CoT Vectors that achieve greater stability and stronger performance. We comprehensively evaluate both CoT Vectors against baseline and LoRA across two models and four benchmarks, and interpret performance variations through analyses of latent space structure, information density, acquisition mechanisms, and pre-training differences. Together, these studies not only extend the task vector framework to reasoning, but also reveal new perspectives on the underlying mechanisms of LLM reasoning.

### 4.2.1 THE THREE-STAGE REASONING PROCESS

To assess whether task vectors can be extended to reasoning, we first explore the applicability of conventional extracted task vectors in CoT setting. Table 1 shows that Extracted CoT Vectors are indeed effective, improving over the baseline by an average of 2.4 and 1.1 points on the two models respectively and demonstrating the feasibility of task vectors in reasoning; however, their effectiveness is highly unstable across different layers (Figure 3 (a)) with the layer-wise average performance even falling below the baseline. Interestingly, this instability follows a non-random pattern. We observe a sawtooth U-shaped pattern: despite the fluctuations, the overall trend shows that performance enhancements when vectors are injected into either the shallow and deep layers, whereas injections into the middle layers yield minimal gains or even degrade performance. This contrasts with prior task vector research on simpler tasks (Todd et al., 2023; Hendel et al., 2023), where middle-layer interventions are typically most effective. This divergence suggests that the functional organization of complex multi-step reasoning in LLMs differs fundamentally from that of simpler tasks, highlighting the unique mechanisms involved in complex reasoning.

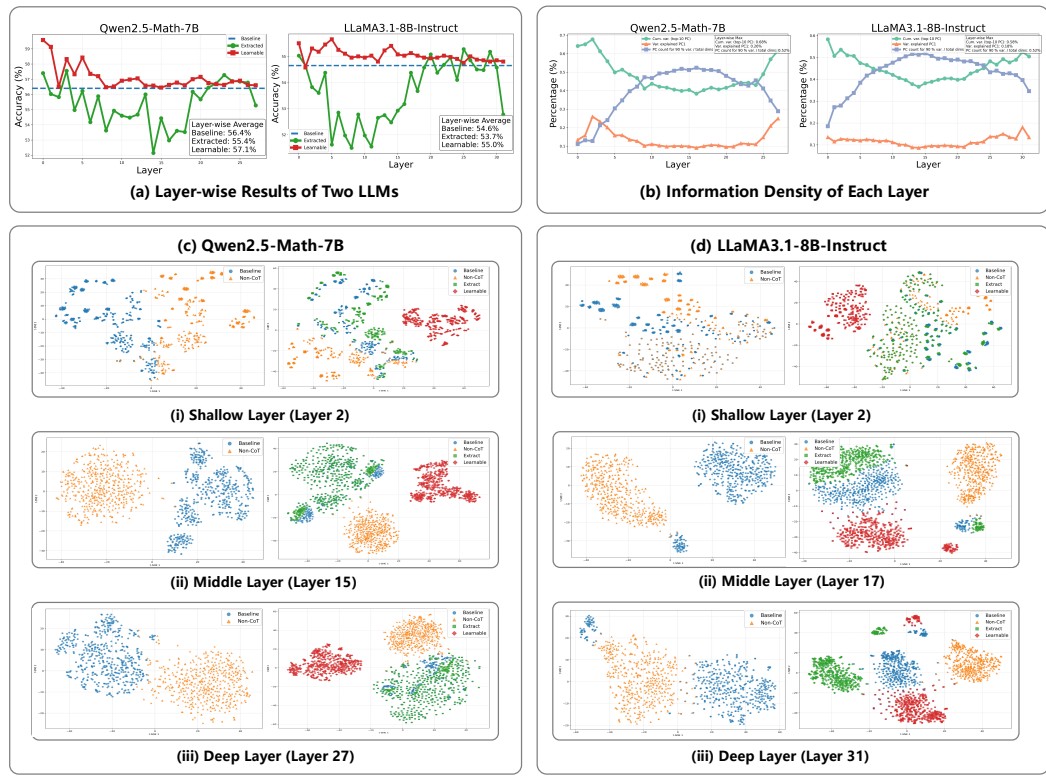

Figure 3: (a) Layer-wise performance of two LLMs with both extracted and Learnable CoT Vectors, averaged over four datasets. (b) Layer-wise information density curves of two LLMs, obtained via PCA on 500 sampled instances across four datasets. Abbreviations: PC = principal component; var. = variance; cum. = cumulative; dims = dimensions. (c–d) T-SNE visualizations of hidden states at shallow, middle, and deep layers on GSM8K (500 samples) of two LLMs. Left: sample distributions under non-CoT and baseline (with CoT) inputs. Right: same baseline with additional insertion of Extracted and Learnable CoT Vectors. Color scheme is consistent across (a, c, d): orange = non-CoT, blue = baseline, green = Extracted CoT Vector, red = Learnable CoT Vector.

This layer-wise variability naturally leads us to hypothesize that the underlying reasoning process in LLMs may itself be structured in stages. Building on insights from prior work on layer specialization (Tenney et al., 2019; Chuang et al., 2023; Skean et al., 2025), we posit a three-stage organization of perception, reasoning, and expression. In this view, Shallow layers primarily perform basic feature extraction and semantic encoding, producing more linear and unified representations. Middle layers execute core reasoning process, leading to sample-specific, high-dimensional representations with no dominant direction. Deep layers map internal reasoning states into surface-level linguistic outputs, where the representations again become more unified.

To test this hypothesis, we conduct an information density analysis via PCA on hidden states from 500 randomly sampled instances (Figure 3 (b)). We observe that mid-layers require significantly more principal components to explain the variance compared to shallow and deep layers, while the variance explained by the top components drops sharply. This indicates higher representational complexity and the absence of a dominant direction in mid-layers, ultimately delineating three distinct stages across shallow, middle, and deep layers. Visualizing the latent space with t-SNE (Figures 3 (c-d)) further reveal that middle-layer activations with CoT (baseline) form dispersed, input-specific clusters, reflecting a highly complex and non-linear structure that differs markedly from the non-CoT distribution. In contrast, shallow and deep layers exhibit more uniform activations, supporting that the middle layers serve as core stage for sample-specific reasoning. These findings explain why Extracted CoT Vectors fail in the middle layers: the mid-layer activations lack a coherent, task-general direction, making it difficult to extract a compact and reusable CoT Vector.

Further cross-layer transfer experiments support this conclusion. In Table 2, injecting mid-layer vectors into shallow layers degrades performance, whereas shallow-layer vectors improve performance

Table 1: Comprehensive evaluation results. MATH-E = MATH-Easy, MATH-H = MATH-Hard, MMLU-P = MMLU-Pro, CSQA = CommonsenseQA, SQA = StrategyQA. Reported CoT Vector results correspond to the best injection layer selected from layer-wise evaluation. We note that extraction-based vectors are particularly dependent on this choice, whereas learnable vectors maintain more consistent performance across layers.

| Model | Method | #Params | GSM8K | MATH-E | MATH-H | MMLU-P | CSQA | SQA | Avg. |
|---|---|---|---|---|---|---|---|---|---|
| Qwen2.5-Math-7B | Baseline | — | 74.6 | 69.9 | 47.9 | 33.2 | 53.8 | 23.7 | 50.5 |
| | Extracted | — | 78.2 | **72.0** | 49.7 | **35.3** | 57.5 | 29.1 | 53.6 |
| | Learnable | 3.6K(×1.0) | **83.5** | 71.9 | **50.9** | 35.1 | **58.2** | **31.2** | **55.1** |
| | LoRA | 10.0M(×2777.8) | 79.0 | 70.4 | 48.2 | 33.8 | 58.0 | **31.2** | 53.4 |
| LLaMA3.1-8B-Instruct | Baseline | — | 77.4 | 62.0 | 34.6 | 44.6 | 72.7 | 60.8 | 58.7 |
| | Extracted | — | **78.6** | 63.2 | 35.7 | 45.5 | 73.2 | 64.3 | 60.1 |
| | Learnable | 4.2K(×1.0) | 78.2 | **63.8** | **36.4** | **46.2** | **73.7** | **65.0** | **60.6** |
| | LoRA | 13.6M(×3238.0) | **78.6** | 63.5 | 36.3 | 45.5 | 73.6 | 64.8 | 60.4 |

Table 2: Cross-layer CoT Vector transfer results on Qwen-GSM8K. Performance when injecting a CoT Vector extracted from a Source Layer (column) into a different Target Layer (row). The diagonal shows baseline performance (source = target). Δ indicates the absolute change from the target layer's baseline. Green arrows (↑) indicate improvement, red arrows (↓) indicate degradation.

| Target Layer | Source: Shallow (L6) | | Source: Middle (L14) | |
|---|---|---|---|---|
| | Accuracy | Δ | Accuracy | Δ |
| Shallow (Layer 6) | 78.2 | — | 63.8 | ↓14.4 |
| Middle (Layer 14) | 75.3 | ↑9.0 | 66.3 | — |

when injected into middle layers. This indicates that the failure of mid-layer injection stems not from location, but from the intrinsically sample-specific and non-generalizable nature of mid-layer representations, which are ill-suited for capturing a compact, task-wide reasoning direction.

### 4.2.2 LEARNABLE CoT VECTORS

**Learnable vs. Extracted CoT Vectors.** Beyond the conventional extraction-based approach, we further introduce novel Learnable CoT Vectors, optimized via a teacher-student architecture to distill generalizable reasoning patterns. Experimental results reveal that the Learnable CoT Vector demonstrates two clear advantages over its extracted counterpart: (i) higher overall performance across benchmarks (Table 1), and (ii) significantly greater stability across layers with higher layer-wise average accuracy (Figure 3 (a)). Unlike the sawtooth U-shaped curve observed for extracted vectors, where gains concentrate in shallow and deep

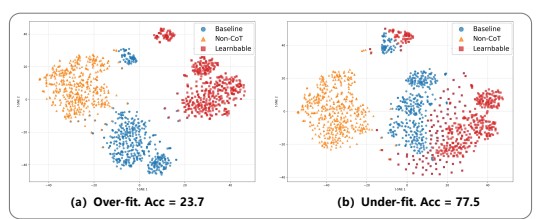

(a) Over-fit. Acc = 23.7    (b) Under-fit. Acc = 77.5

Figure 4: T-SNE visualization of over-fit and under-fit Learnable CoT Vectors (Layer 30 of LLaMA on GSM8K).

layers but diminish in the middle and with strong fluctuations, the learnable vector peaks at the first layers and maintains a consistent plateau across all subsequent layers. Consequently, while extracted vectors show noticeable drops at mid-layers compared to baseline, learnable vectors consistently provide improvements across nearly all layers.

We attribute this divergence to the fundamental nature of each vector type. The extracted vector is a descriptive statistic, passively recording the average activation difference between CoT and non-CoT forward passes. Its efficacy is thus constrained by the representational properties of the source layer: strong when representations have clear dominant directions (e.g., shallow and deep layers), but fragile when such structure is absent. As a consequence, it not only induces relatively mild shifts in latent space (Figure 3 (c-d)) but also retains sample-specific noise, leading to sharp volatility where even adjacent layers at similar depths behave inconsistently. In contrast, learnable

vectors are optimized via gradient descent to mimic the teacher model's reasoning. This results in a more directional and aggressive shift in the latent space (Figure 3 (c-d)), enabling it to overcome representational limitations of individual layers and avoid being intervened by sample-specific noise. Consequently, the learnable vector achieves stronger and more stable performance across layers.

These differences have important practical implications. Extracted vectors suffer from high instability, with the optimal injection layer varying across tasks and models (see Appendix for more details). In real-world deployment, where ground truth is unavailable, such unpredictability severely limits their usability. Learnable CoT Vectors, however, produce consistent gains across all layers, with their strongest performance consistently emerging in the shallowest layers—often at the very first layer. This stability permits simple and robust application: even on unseen tasks, injecting the vector at the first layer suffices to achieve near-optimal performance.

However, the aggressive steering of Learnable CoT Vectors also brings risks. As visualized in Figure 4, vectors applied to middle or deep layers are prone to overfitting, over-steering the latent space and collapsing diverse reasoning paths, which significantly degrades accuracy. This fragility stems from the representational nature of these layers that mid-layer activations are heterogeneous and sample-specific, while deep layers are closely tied to surface outputs, where even small perturbations can destabilize generation. To mitigate this, we employ early stopping or reduced learning rates, which produce mildly under-fitted vectors that still provide modest gains without catastrophic collapse. These findings reinforce our earlier conclusion that shallow layers are the most suitable for Learnable CoT Vectors, while mid and deep layers are less amenable to strong external guidance.

**Learnable CoT Vectors vs. LoRA.** From parameter-efficiency perspective, our Learnable CoT Vector demonstrates advantages over LoRA fine-tuning. As illustrated in Table 1, it outperforms LoRA on most datasets while requiring orders of magnitude fewer trainable parameters. We attribute this to the fact that instruction-tuned LLMs already possess strong CoT priors, leaving limited room for LoRA to improve. In contrast, our approach adds an external guidance signal that efficiently steers the model's latent reasoning without altering the model's existing functional structure.

### 4.2.3 MODEL DIFFERENCES

As shown in Table 1, the effectiveness of CoT Vectors varies across models: Qwen benefits more consistently and substantially from CoT Vector injection compared to LLaMA. For example, averaged across benchmarks, Qwen gains up to 4 points over the baseline, whereas LLaMA yields a more modest improvement of 1.5 points with Learnable CoT Vectors. We trace this discrepancy to differences in the latent space structures of the two models throughout the three-stage reasoning process. In Figure 3 (b), Qwen exhibits a more distinct three-stage reasoning pattern than LLaMA. Notably, its top principal components explain more variance than those of LLaMA, suggesting a lower information density and a more structured latent space with clearer principal directions. This facilitates both extraction and optimization in capturing high-quality, task-general signals.

We conjecture that this structural disparity stems from differences in training data and procedures. Qwen has undergone more domain-focused and standardized fine-tuning, whereas LLaMA has been trained on broader and less curated corpora. As a result, Qwen demonstrate a more distinct functional separation of layers. This structural clarity allows CoT Vectors to more easily capture task-general reasoning directions. In summary, the performance gap highlights that the efficacy of CoT Vectors is influenced by the inherent properties of the model's representations. Models with more structured latent spaces provide a more fertile ground for the CoT Vector intervention.

### 4.2.4 CROSS-DATASET AND CROSS-MODEL TRANSFERABILITY

We investigate whether CoT Vectors acquired from one source (model or dataset) can be effectively applied to another.

**Cross-Model Transfer.** As shown in Table 3, CoT Vectors gained from one model can be effectively reused in another. The vectors obtained from more powerfully instruction-tuned variant of the Qwen2.5 series (Qwen2.5-Math-7B-Instruct) consistently improve performance when applied to Qwen2.5-Math-7B (74.6 $\rightarrow$ 77.5).

**Cross-Dataset Transfer.** Results in Table 3 further demonstrate transferability across datasets. 1) In-domain: CoT Vectors obtained from the GSM8K dataset effectively enhance performance on the

| Source → Target | Baseline | Self | Transferred |
|---|---|---|---|
| *Cross-Model Transfer* | | | |
| Qwen2.5-Math-7B-Instruct → Qwen2.5-Math-7B | 74.6 | 78.2 | 77.5 |
| *Cross-Dataset Transfer* | | | |
| GSM8K → MATH | 47.9 | 49.7 | 48.6 |
| MMLU-Pro → MATH | 47.9 | 49.7 | 48.5 |

Table 3: Cross-model and Cross-dataset transfer results of CoT Vectors. Baseline refers to standard zero-shot CoT prompting. Self means applying the CoT Vector obtained from the same model–dataset pair (no transfer). Transferred means applying a CoT Vector obtained from a different source model or dataset.

MATH dataset ($47.9 \rightarrow 48.6$). This confirms that the vector successfully captures a generalized mathematical reasoning strategy rather than merely memorizing dataset-specific features. 2) Cross-domain: vectors obtained from MMLU-Pro yield gains on MATH ($47.9 \rightarrow 48.5$). This suggests that the CoT Vector may encode a meta-reasoning capability—such as the ability to decompose problems or follow logical steps—that is beneficial across distinct task domains.

These transferability experiments underscore a central claim of our work: the CoT Vector is not merely a compressed set of features from a specific model or dataset, but a portable, generalizable representation of a reasoning process that can be effectively applied in novel contexts.

### 4.2.5 ABLATION ON TRAINING SET SIZE FOR LEARNABLE CoT VECTORS

We further conduct an ablation study on the size of the support set to compare the performance of the Learnable CoT Vector and LoRA under different data regimes. As shown in Table 4, while both methods benefit from larger support sets, the Learnable CoT Vector consistently outperforms LoRA across all data scales. Notably, with a very small support set (e.g., 100 examples), the learnable CoT Vector still yields noticeable improvements over the baseline, whereas LoRA offers only marginal gains. This highlights the strong data efficiency of our approach. As the support set grows, Learnable CoT Vector also demonstrates greater potential for performance improvement compared to LoRA. These phenomena all indicate that our Learnable CoT Vectors provide a more effective and scalable mechanism for enhancing reasoning performance than LoRA across diverse data conditions.

Table 4: Performance Comparison with Different Training Sample Sizes on Qwen-GSM8K.

| Sample Size | Baseline | Learnable CoT Vector | LoRA |
|---|---|---|---|
| 100 | 74.6 | **78.2** | 76.0 |
| 500 | 74.6 | **79.0** | 77.9 |
| 1000 | 74.6 | **82.3** | 78.5 |
| 3000 | 74.6 | **83.5** | 79.0 |

Overall, our results confirm that CoT Vectors are a highly efficient and effective means of enhancing reasoning capabilities. Due to space limitations, further ablation studies and robustness analyses of CoT Vectors are provided in the supplementary material.

## 5 CONCLUSION

We have presented CoT Vectors, extending the task vector paradigm to multi-step reasoning in LLMs. Our analyses uncover a consistent three-stage reasoning process and show that the newly introduced Learnable CoT Vectors provide stronger and more stable gains than the traditional extraction-based approach, while also offering multiple perspectives on why their effectiveness differs. These results demonstrate both the practical utility of CoT Vectors and their value as a probe into the mechanisms and organization of multi-step reasoning in LLMs. However, performance variability in intermediate layers highlights structural limitations, suggesting that task-level vectors may not fully capture intra-task diversity. Future work could explore finer-grained or adaptive vectorization strategies to improve robustness and generalization.

ACKNOWLEDGMENTS

This work is supported by Jiangsu Province Carbon Peak Carbon Neutrality Science and Technology Innovation Special Fund Project (Grant No. BT2025029), National Natural Science Foundation of China (62576091), the Southeast University Big Data Computing Center, and Southeast University Kunpeng & AscendCenter of Cultivation.

ETHICS STATEMENT

This work adheres to the ICLR Code of Ethics. Our study does not involve human subjects, personally identifiable information, or proprietary data. All datasets used, including GSM8K, MATH, and MMLU-Pro, are publicly available. The proposed method, CoT Vectors, is a parameter-efficient technique for steering the reasoning process of pre-trained large language models. It does not introduce any new capabilities that could cause harm, nor does it enable misuse beyond the standard capabilities of existing large language models. We are not aware of any potential risks related to bias, fairness, or security that arise specifically from the method proposed. However, we acknowledge that the effectiveness and potential output of CoT Vectors are dependent on the base model and the support set data; as such, they may reflect or amplify biases present in these sources. No conflicts of interest, legal compliance issues, or sponsorship-related influences are present in this work.

REPRODUCIBILITY STATEMENT

We have taken multiple steps to ensure the reproducibility of our work. All datasets used in our experiments are publicly available and properly cited in the main text and appendix. Training configurations, including hyperparameters, optimizers, and evaluation settings, are described in detail in Section 4.1 and Appendix A.3. Theoretical claims, including the formalization of the CoT shift, are formally derived in Section 3.1 and Appendix A.2. Experimental results include multiple models, reasoning benchmarks, and various ablations to validate robustness in Section 4 and Appendix A.4-A.5. We will release the full source code and pre-trained vectors upon publication to further support reproducibility.

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

# A APPENDIX

## A.1 THE USE OF LARGE LANGUAGE MODELS

In preparing this manuscript, we use Large Language Models (LLMs) solely as a writing assistant for grammar correction and minor phrasing refinement. The LLMs doesn't contribute to research ideation, experimental design, analysis, or interpretation of results. All scientific content and claims are entirely the responsibility of the authors. The LLMs are not involved in authorship and are not considered contributors to this work.

## A.2 THE DERIVATION OF CoT SHIFT

When the model reasons with CoT, the generation of an answer token $a \in A$ depends not only on the question tokens but also on the intermediate CoT tokens. Formally, the single-head self-attention can be expressed as

$$\text{SA}(a, [\boldsymbol{K}_Q, \boldsymbol{K}_C, \boldsymbol{K}_A], [\boldsymbol{V}_Q, \boldsymbol{V}_C, \boldsymbol{V}_A]) = \frac{\exp(a\boldsymbol{K}_Q^\top)}{Z_{\text{total}}}\boldsymbol{V}_Q + \frac{\exp(a\boldsymbol{K}_C^\top)}{Z_{\text{total}}}\boldsymbol{V}_C + \frac{\exp(a\boldsymbol{K}_A^\top)}{Z_{\text{total}}}\boldsymbol{V}_A$$

$$= \frac{Z_Q}{Z_{\text{total}}} \cdot \underbrace{\frac{\exp(a\boldsymbol{K}_Q^\top)}{Z_Q}\boldsymbol{V}_Q}_{\text{SA}(a,[\boldsymbol{K}_Q],[\boldsymbol{V}_Q])} + \frac{Z_C}{Z_{\text{total}}} \cdot \underbrace{\frac{\exp(a\boldsymbol{K}_C^\top)}{Z_C}\boldsymbol{V}_C}_{\text{SA}(a,[\boldsymbol{K}_C],[\boldsymbol{V}_C])} + \frac{Z_A}{Z_{\text{total}}} \cdot \underbrace{\frac{\exp(a\boldsymbol{K}_A^\top)}{Z_A}\boldsymbol{V}_A}_{\text{SA}(a,[\boldsymbol{K}_A],[\boldsymbol{V}_A])}$$

where $Z_Q = \sum \exp(a\boldsymbol{K}_Q^\top)$, $Z_C = \sum \exp(a\boldsymbol{K}_C^\top)$, $Z_A = \sum \exp(a\boldsymbol{K}_A^\top)$, and $Z_{\text{total}} = Z_Q + Z_C + Z_A$.

In contrast, without CoT, the self-attention reduces to

$$\text{SA}(a, [\boldsymbol{K}_Q, \boldsymbol{K}_A], [\boldsymbol{V}_Q, \boldsymbol{V}_A]) = \frac{Z_Q}{Z_Q + Z_A} \cdot \text{SA}(a, [\boldsymbol{K}_Q], [\boldsymbol{V}_Q]) + \frac{Z_A}{Z_Q + Z_A} \cdot \text{SA}(a, [\boldsymbol{K}_A], [\boldsymbol{V}_A])$$

Therefore, through algebraic transformations and let $\mu = \dfrac{Z_C}{Z_{\text{total}}}$, we can obtain,

$$\text{SA}(a, [\boldsymbol{K}_Q, \boldsymbol{K}_C, \boldsymbol{K}_A], [\boldsymbol{V}_Q, \boldsymbol{V}_C, \boldsymbol{V}_A]) = \underbrace{\text{SA}(a, [\boldsymbol{K}_Q, \boldsymbol{K}_A], [\boldsymbol{V}_Q, \boldsymbol{V}_A])}_{\text{Standard attention}} \tag{8}$$

$$+ \mu \cdot \underbrace{(\text{SA}(a, [\boldsymbol{K}_C], [\boldsymbol{V}_C]) - \text{SA}(a, [\boldsymbol{K}_Q, \boldsymbol{K}_A], [\boldsymbol{V}_Q, \boldsymbol{V}_A]))}_{\text{CoT shift}}$$

$$\tag{9}$$

This is Equation 1 in the main text, which reveals that when LLM performs CoT inference, the actual content of CoT is a prefix for the final answer given. Its role in inference ultimately comes from an attention shift, which means that CoT information can be compressed into this shift. This discovery provides a theoretical basis for CoT Vectors.

## A.3 MORE IMPLEMENTATION DETAILS

### A.3.1 DATASET DETAILS

**GSM8K.** We randomly sample 3,000 examples from the training set to construct the support set for CoT vector extraction/learning and LoRA fine-tuning. For evaluation, we randomly selected 1,000 examples from the test set.

**MATH.** We have observed that the MATH dataset contains problems with five levels of difficulty, and the imbalanced distribution across these levels could lead to unstable training. To address this, we partition the dataset into two subsets based on difficulty: MATH-Easy (levels 1–3) and MATH-Hard (levels 4–5). For each subset, we combine all categories and then sample 3,000 examples from the training portion to form the support set. Similarly, 1,000 examples were sampled from the test portion for evaluation.

**MMLU-Pro.** Since ground-truth CoT sequences are required, we have used the official validation set as the support set. Although this set contains only 70 samples, it still provides sufficient signal for CoT vectors to capture useful reasoning patterns. For testing, we also sample 1,000 examples from the test set.

### A.3.2 PROMPTS AND GENERATION CONFIGURATIONS

**Prompts.** We use two versions of prompts: CoT and Non-CoT, as shown in Table 5. The CoT version is specifically designed for scenarios that require step-by-step reasoning. This prompt format is consistently applied to all our baseline model tests and to the evaluation processes for both the CoT Vectors and LoRA methods. It also serves as the teacher input during the training of our Learnable CoT Vectors. Conversely, the Non-CoT version is used to directly obtain the final answer without any intermediate reasoning, serving exclusively as the student input during the training of our Learnable CoT Vectors.

Additionally, for GSM8K and MATH datasets, we instruct the LLM to directly generate the final answer, while for the multiple-choice dataset MMLU-Pro, we prompt the model to output the correct option letter. In the input construction for MMLU-Pro, the answer choices are appended immediately after the question.

Table 5: Prompt Templates.

| Dataset | Reasoning Type | Prompt |
|---|---|---|
| GSM8K & MATH | CoT | You are a helpful and precise assistant for solving math problems. Please reason step by step, and put your final answer within \boxed{}. |
| | non-CoT | You are a helpful and precise assistant for solving math problems. Put your answer within \boxed{}. |
| MMLU-Pro | CoT | You are a helpful and precise assistant for solving problems. Please reason step by step, and put your final answer within \boxed{}. Your final output should be only the uppercase letter of the correct choice (e.g., A). |
| | non-CoT | You are a helpful and precise assistant for solving problems. Put your answer within \boxed{}. Your final output should be only the uppercase letter of the correct choice (e.g., A). |

**Generation Configurations.** All tests were conducted using a consistent set of generation parameters, as detailed in Table 6.

Table 6: Generation Configuration Parameters

| Parameter | Value |
|---|---|
| Number of beams | 3 |
| Maximum new tokens | 512 |
| Length penalty | 0.0 |
| Do sample | False |
| Temperature | 1.0 |
| Top-p | 1.0 |

### A.3.3 HYPERPARAMETERS

We employee the AdamW optimizer (Loshchilov & Hutter, 2017) with a weight decay of 1e-3 and train using float16 precision for both Learnable CoT Vector and LoRA. Additionally, we set the accumulate gradient batches to 2 to effectively increase the batch size.

**Learnable CoT Vectors.** To train our Learnable CoT Vectors, we configure distinct hyperparameters for each model and dataset. All of these parameters are fine-tuned through a combination of systematic grid search and manual adjustments to ensure stable and efficient training. For the LLM, we adopt a unique tiered learning rate strategy. Using higher learning rate for the earlier layers of Qwen is intended to obtain vectors that better fit the task-general direction, as mentioned in Section 4.2 of the main text, where shallow layers exhibit better directional properties. In contrast, using smaller learning rates for the deeper layers of Qwen and the entire Llama model aims to obtain underfitting vectors, as they demonstrate better stability when information density is high. As detailed in Table 7.

Table 7: Hyperparameters for Learnable CoT Vector

(a) Qwen

| Dataset | Samples | Learning Rate | Warm-up | Note |
|---------|---------|---------------|---------|------|
| GSM8K   | 3000    | 5e-3 and 1e-4 | 0.1     | LR: First 4 layers used 5e-3, others 1e-4 |
| MMLU-P  | 70      | 5e-3 and 1e-4 | 0.1     | LR: First 4 layers used 5e-3, others 1e-4 |
| MATH-E  | 2000    | 5e-3 and 1e-4 | 0.5     | LR: First 4 layers used 5e-3, others 1e-4 |
| MATH-H  | 2000    | 5e-3 and 1e-4 | 0.5     | LR: First 4 layers used 5e-3, others 1e-4 |

(b) LLaMA

| Dataset | Samples | Learning Rate | Warm-up |
|---------|---------|---------------|---------|
| GSM8K   | 2000    | 1e-4          | 0.5     |
| MMLU-P  | 70      | 1e-4          | 0.1     |
| MATH-E  | 1000    | 1e-4          | 0.5     |
| MATH-H  | 1000    | 1e-4          | 0.5     |

**LoRA.** We apply LoRA with a set of shared, general hyperparameters across all models and datasets, with training set sizes consistent with those used for CoT vector training. Interestingly, we found that when using LoRA for fine-tuning, the loss was very low from the beginning of training. This aligns with our hypothesis in Section 4.2 that for models already fine-tuned with CoT and possessing CoT capabilities, using LoRA for further specific CoT fine-tuning would yield limited gains. As detailed in Table 8.

Table 8: General Hyperparameters for LoRA.

| Parameter | Value |
|-----------|-------|
| Learning Rate | $5 \times 10^{-5}$ |
| Warm-up | 0.1 |
| Matrix Rank | 16 |
| Dropout Rate | 0.05 |
| Target Modules | q_proj, k_proj, o_proj, v_proj |

### A.4 COMPLETE EXPERIMENTAL RESULTS

In Section 4.2 of the main text, we provide the comprehensive evaluation results for two types of CoT Vectors, baseline and LoRA cross models and benchmarks, in Table 1. Especially for CoT Vectors, the final result is the report of the best layer after iterating and testing the results of each layer of LLMs. Here we present all the layer-wise results of each model and dataset in Figure 5.

We can observe the following pattern. First, as mentioned in the main text, extracted layers generally exhibit performance troughs in the middle layer, with strong volatility. Relatively speaking, learnable layers usually achieve their best performance in the first few layers, while the subsequent layers are generally stable around the baseline or with a small gain around the baseline. Second, in addition, the most crucial point is that for highly volatile extracted vectors, we find that even if

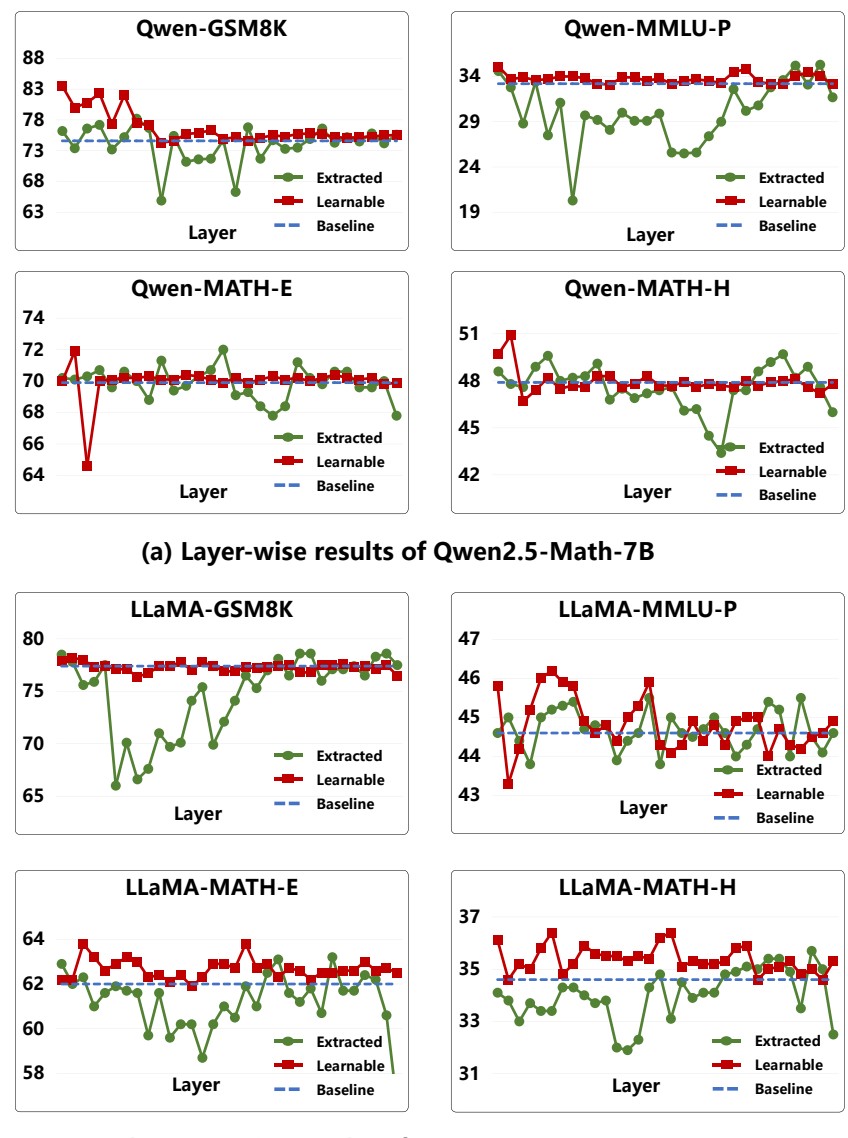

(a) Layer-wise results of Qwen2.5-Math-7B

(b) Layer-wise results of LLaMA3.1-8B-Instruct

Figure 5: Layer-wise performance analysis of two LLMs and four benchmarks.

the same model is used, the layers that achieve the best performance on different datasets are completely different. Even a certain layer can gain a lot on one dataset, but it will damage performance on another dataset, and there is basically no regularity in this. This means that when using extracted CoT Vectors, a layer wise search must be conducted first to determine the optimal shift layer, which is almost impractical for practical applications without ground truth.

## A.5 ABLATIONS AND ADDITIONAL ANALYSES

Due to space limitations in the main paper, we provide extended analyses of CoT Vectors in this section. We first present alternative acquisition methods beyond the activation-gap based approach, including raw-activation (Section A.5.1) and attention-level variants (Section A.5.2), which also improve over the baseline but yield less representative gains than the method reported in the main text. We then conduct ablation studies on key design choices, such as the scaling factor $\mu$ for extracted vectors (Section A.5.4), the loss balance for learnable vectors (Section A.5.6), and the training set size (Section 4.2.5), through which we uncover additional insights into their behaviors. We fur-

Table 9: Results of Raw Activation Extracted CoT Vectors.

| Model | Method | #Params | GSM8K | MATH-E | MATH-H | MMLU-P | Avg. |
|---|---|---|---|---|---|---|---|
| | Baseline | — | 74.6 | 69.9 | 47.9 | 33.2 | 56.4 |
| Qwen2.5-Math-7B | Raw Activation | — | 77.6 | 71.5 | 49.2 | **35.8** | 58.5 |
| | Activation Gap | — | **78.2** | **72.0** | **49.7** | 35.3 | **58.8** |
| | Baseline | — | 77.4 | 62.0 | 34.6 | 44.6 | 54.7 |
| LLaMA3.1-8B-Instruct | Raw Activation | — | **80.4** | 62.4 | 35.4 | **45.8** | **56.0** |
| | Activation Gap | — | 78.6 | **63.2** | **35.7** | 45.5 | 55.8 |

ther analyze robustness and transferability, considering settings such as multi-layer vector injection (Section A.5.3), varying the source of teacher CoT sequences (Section A.5.5), and cross-model or cross-dataset transfer (Section 4.2.4). Finally, we complement these quantitative studies with case analyses (Section A.5.7), illustrating both successful and failure scenarios to better understand when and why CoT Vectors are effective.

### A.5.1 RAW ACTIVATION EXTRACTED COT VECTOR

This method is inspired by the work in ICL (Todd et al., 2023; Hendel et al., 2023), using the direct model activations as a coarse-grained approximation of the CoT Vector. Specifically, we forward-pass the triplet $(Q, \text{CoT}, A)$ and take the mean of the hidden states (activations) across all answer tokens $A$ at a specific layer $l$.

$$\vec{v}_{\text{CoT}}^{(l)} = \frac{1}{|A|} \sum_{a \in A} \boldsymbol{\alpha}_{\text{CoT}}^{(l)}(a) \tag{10}$$

where $\boldsymbol{\alpha}_{\text{CoT}}^{(l)}(a)$ is the hidden state (activation vector) of the answer token $a$ at layer $l$ under the CoT prompting input, and $|A|$ denotes the total number of answer tokens.

**Raw Activation vs. Activation Gap.** Comparing two extraction strategies in Table 9, the Activation-Gap vector generally outperforms the Raw-Activation vector in most datasets, as it better isolates the reasoning-induced shift. Interestingly, in some cases raw activations match or even surpass gap-based ones, suggesting that hidden states encode not only the reasoning information but also auxiliary task-relevant features.

### A.5.2 ATTENTION-LEVEL COT VECTOR

From a mathematical perspective, the most precise location for the CoT Vector should be within the attention computation, as shown in Eq. 3 in main text. Therefore, in addition to the Activation-level CoT Vector presented in the main text, we have also explored an intervention at the attention level in both Extracted and Learnable CoT Vectors. This method similarly improve performance over the baseline but is slightly less effective than the activation-level approach.

**Attention-level Extracted CoT Vector.** We extract the attention output at a specific layer when the LLM processes the (Q, CoT, A) triplet and the (Q, A) pair separately during the forward pass, and compute the gap between them as a highly accurate attention-output-level CoT Vector. For multi-head attention, we compute a distinct vector $\vec{v}_{\text{CoT, head}}^{(i,h)}$ for each head $h$, enabling more fine-grained control. The final task vector is a collection of all these head-level vectors. The extracted CoT Vector in activation form is inserted into the hidden states of the corresponding extraction layer, while the attention-gap form is inserted into the attention output of that same layer. During insertion, a manually set constant $\mu$ is used to scale the magnitude of the CoT shift.

**Attention-level Learnable CoT Vector.** Similar to the activation level learnable method, but the attention-level vector is added as a shift to the output of a specific attention module. For multi-head attention, a distinct vector is learned for each head, allowing for fine-grained control.

In practice, although this method does lead to some performance improvement against baseline, the gains are relatively modest compared to activation-level variants. The details of these experiments

Table 10: Results of Attention-level CoT Vectors.

| Model | Method | #Params | GSM8K | MATH-E | MATH-H | MMLU-P | Avg. |
|---|---|---|---|---|---|---|---|
| | Baseline | — | 74.6 | 69.9 | 47.9 | 33.2 | 56.4 |
| Qwen2.5-Math-7B | Extracted | — | 76.8 | 71.1 | 49.7 | 35.2 | 58.2 |
| | Learnable | 7.2K | 76.1 | 71.9 | 48.6 | 34.1 | 57.9 |
| | Baseline | — | 77.4 | 62.0 | 34.6 | 44.6 | 54.7 |
| LLaMA3.1-8B-Instruct | Extracted | — | 78.3 | 62.7 | 35.4 | 46.2 | 55.7 |

are listed in Table 10. This result suggests that the effectiveness of task-general CoT Vectors does not rely on fine-grained, head-specific attention patterns, but instead reflects broader representational shifts at the activation level.

### A.5.3 MULTI-LAYER COT INJECTION

In our main experiments, our designed CoT Vector is exclusively obtained and inserted at a single layer of the LLM. However, we are also curious about whether such operations could be applied simultaneously across multiple layers, and whether performing shifts in more layers would lead to greater performance improvements.

**Extracted CoT Vectors.** As shown in Table 11, we find that simultaneously injecting the extracted CoT Vectors into multiple layers produces different effects, depending on the functional characteristics of the selected layers. When vectors are injected into multiple layers that prove beneficial for CoT integration(e.g. Layer 3, 7, 15 for GSM8K), a positive synergistic effect is observed, resulting in greater performance improvement than single-layer injection. Conversely, when vectors are simultaneously inserted into multiple layers that are sensitive to CoT but unsuitable for external intervention(e.g. Layer 8, 14), additional interference effects arise, leading to a further decline in performance,which sometimes even significantly below the baseline. This phenomenon reaffirms the significant functional specialization among different layers of the Transformer and also indicates that the injection effect of CoT Vectors can be further optimized through inter-layer coordination and combinatorial selection.

Table 11: Multi-layer Extracted CoT Injection on Qwen-GSM8K.

| Model | Layers | Accuracy (%) |
|---|---|---|
| Baseline | – | 74.6 |
| | 15 | 77.6 |
| | 7, 15 | 79.3 |
| Extracted | 3, 7, 15 | **80.1** |
| | 8, 14 | 45.1 |

**Learnable CoT Vectors.** We further investigate multi-layer insertion of Learnable CoT shifts. Interestingly, multi-layer insertion produces complementary effects for extracted vs. learnable vectors: extracted vectors benefit from multi-layer insertion (cumulative mild shifts yield larger gains), while learnable vectors typically perform best when injected at a single shallow layer and degrade under naive multi-layer injection as shown in Table 12. Mechanistically, this difference stems from the amplitude and directionality of the shifts. Extracted vectors produce mild, descriptive shifts that safely accumulate across layers, whereas learnable vectors induce stronger, optimization-driven shifts that can over-steer representations if applied repeatedly.

### A.5.4 ABLATION ON THE SCALING FACTOR $\mu$ OF EXTRACTED COT VECTORS

In the main experiments, the scaling factor $\mu$ for the Extracted CoT Vectors is set to a fixed value of 1.0. To further investigate the impact of this hyperparameter, we experiment with different values of $\mu$.

Table 12: Multi-layer Learnable CoT Injection on Qwen-GSM8K.

| Method | Layers | Accuracy (%) |
|---|---|---|
| Baseline | – | 74.6 |
| Learnable | 0 | **83.5** |
| | 0, 1 | 81.0 |
| | 0, 1, 2 | 76.1 |

**For single-layer injection.** In Table 13, we find that the effect of the scaling coefficient $\mu$ varies significantly across different model layers, leading to the following observations: For layers where $\mu = 1.0$ already yields improvement (e.g., Qwen on GSM8K at layer 6), increasing $\mu$ can further enhance performance slightly, although an excessively large $\mu$ leads to over-steering and performance degradation. In contrast, for layers where $\mu = 1.0$ causes performance degradation (e.g., Qwen on MATH-Hard at layer 17), reducing $\mu$ can mitigate the negative effect, but no value of $\mu$ can bring improvement over the baseline. This indicates that the CoT Vectors extracted from these layers are inherently low-quality, and their deficiency cannot be remedied simply by adjusting $\mu$.

Table 13: Model Accuracy Comparison Across Different $\mu$ Values for single-layer injection

(a) Qwen-MATH-Hard-Layer 17

| $\mu$ | Accuracy (%) |
|---|---|
| 0.0 | **47.9** |
| 0.2 | 46.6 |
| 0.5 | 46.7 |
| 1.0 | 45.8 |
| 1.2 | 44.4 |

(b) Qwen-GSM8K-Layer 6

| $\mu$ | Accuracy (%) |
|---|---|
| 0.0 | 74.6 |
| 0.8 | 78.1 |
| 1.0 | 78.2 |
| 1.2 | **78.4** |
| 1.5 | 78.0 |

**For multi-layer injection.** For many layers of dense shift insertion, it is necessary to reduce the $\mu$ value. As shown in Table 14, when applying CoT Vector shifts to four layers of the LLM concurrently, scaling down $\mu$ to 0.5 leads to further performance improvement. When injecting CoT Vectors into all layers simultaneously, reducing $\mu$ to 0.2 is necessary to achieve optimal results.

Table 14: Model Accuracy Comparison with Different $\mu$ Values for multi-layer injection

(a) Qwen-GSM8K-all layers

| Setting | Accuracy (%) |
|---|---|
| Baseline ($\mu = 0$) | 74.6 |
| $\mu = 0.05$ | 74.8 |
| $\mu = 0.1$ | 76.3 |
| $\mu = 0.2$ | **76.7** |
| $\mu = 0.3$ | 67.1 |
| $\mu = 0.5$ | 14.5 |
| $\mu = 1.0$ | 15.2 |

(b) Qwen-GSM8K

| Model | $\mu$ | Accuracy (%) |
|---|---|---|
| Baseline | - | 74.6 |
| Layers 3,7,15,23 | 1.0 | 79.6 |
| Layers 3,7,15,23 | 0.5 | **80.3** |

### A.5.5 SOURCES OF TEACHER COT

For the explicit CoT sequences used in extraction or as the teacher signal in the learning process, we experiment with different types of sources. These include using ground-truth CoT from the dataset (typically human-written) as well as model-generated CoT sequences, obtained from the model itself.

As shown in Figure 6,experimental results indicate that the performance difference between using ground-truth CoT and model-generated CoT is relatively minor. Both approaches yield comparable

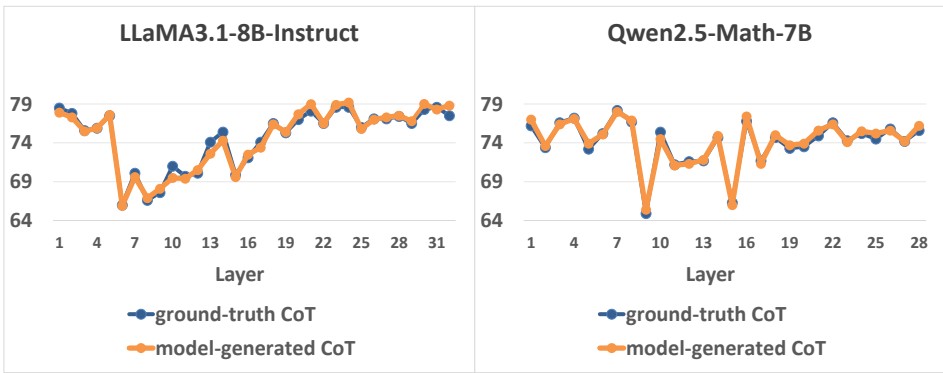

Figure 6: Performance comparison between ground-truth and model-generated CoT sequences of Extracted CoT Vectors on GSM8K.

improvements in the model's reasoning capabilities, suggesting that the benefits of CoT guidance are robust to variations in the specific wording or stylistic differences of the reasoning chains.

### A.5.6 ABLATION ON TRAINING OBJECTIVES FOR LEARNABLE CoT VECTORS

For training the Learnable CoT Vectors, we perform an ablation study on the loss design. In addition to the hybrid loss (a combination of representation alignment loss and cross-entropy loss) adopted in the main experiments, we evaluate two simplified variants: using only the **alignment loss** or only the **cross-entropy loss**. As shown in Table 15, both simplified variants lead to clear performance degradation compared to the hybrid objective.

We attribute this to the complementary roles of the two components. The representation alignment loss is crucial for distilling task-general reasoning patterns: by enforcing the student's intermediate representations to align with the teacher's reasoning trajectory, it effectively transfers the underlying logical process. However, when used alone, it lacks explicit guidance for producing correct and coherent final answers. The cross-entropy loss complements this by anchoring the model's output to accurate reasoning traces, ensuring that the learned vector supports both faithful reasoning dynamics and correct task execution.

Table 15: Ablation Results on Training Objectives in the Qwen–GSM8K Setting

| Method | Accuracy (%) |
|---|---|
| Baseline | 74.6 |
| Learnable CoT Vector | **83.5** |
| Learnable CoT Vector (only alignment loss) | 82.9 |
| Learnable CoT Vector (only CE loss) | 78.4 |

### A.5.7 CASE STUDIES

To qualitatively understand the mechanistic impact of CoT Vectors, we present two representative cases. Figure 7 (a) shows a GSM8K problem where baseline (Zero-Shot CoT) fails. The model's reasoning becomes stuck in an infinite loop, repetitively counting the bids without making progress toward a solution. In contrast, both the Extracted and Learnable CoT Vector helps the model to structure the problem correctly and produce a valid solution. Notably, the learnable vector not only leads to the correct final answer but also produces a reasoning chain that was more structured and closer to the ground-truth CoT in the dataset. This highlights how CoT Vectors can reshape the model's internal reasoning trajectory, guiding it toward more faithful and efficient solutions.

Conversely, we also observe cases where CoT Vectors harm performance. In Figure 7 (b), the baseline arrives at the correct answer unaided. However, when CoT Vectors are injected, the reasoning becomes more aligned with the ground-truth style (especially for extracted vectors), but both vari-

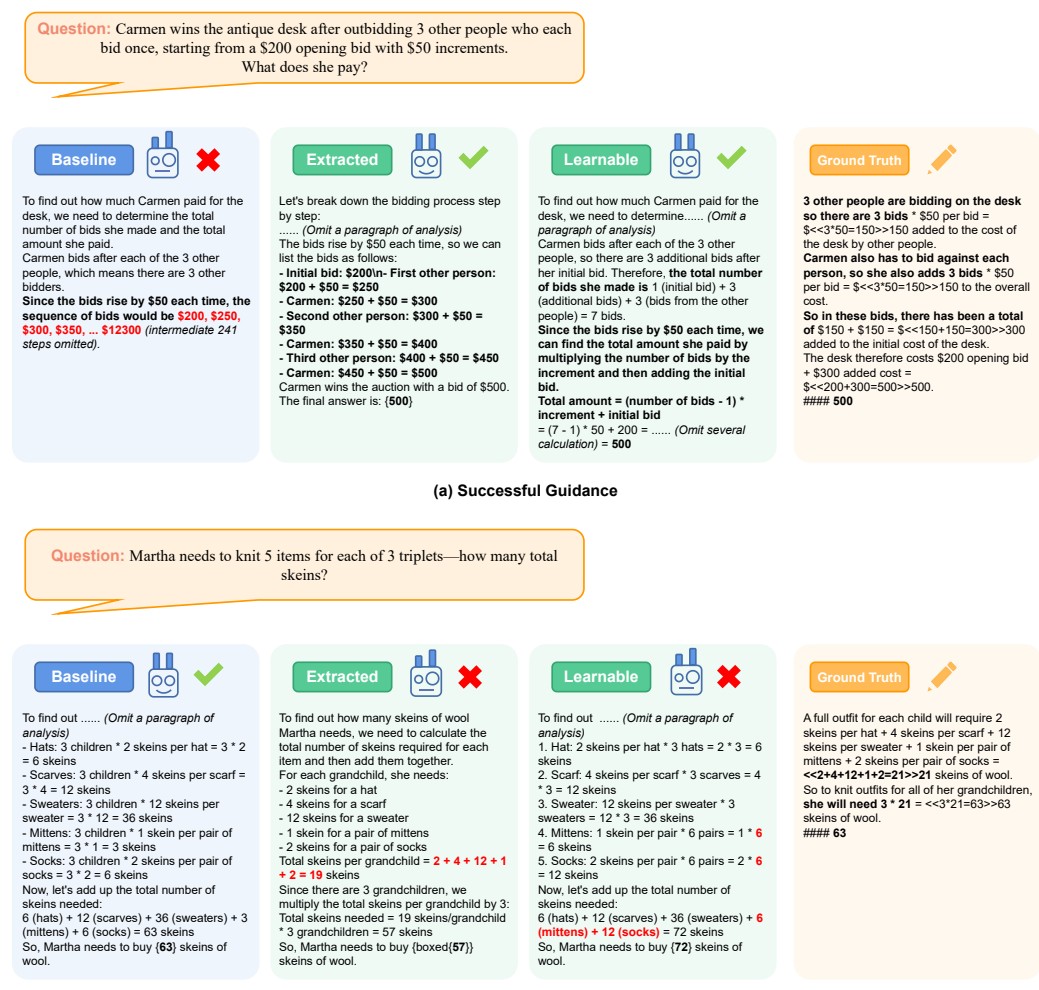

Figure 7: Case studies.

ants introduce hallucinations or arithmetic mistakes that ultimately lead to incorrect conclusions. This demonstrates that while CoT Vectors can effectively shift latent reasoning, the perturbation may sometimes interfere with correct reasoning paths, underscoring the double-edged nature of such interventions.

