# OpenReview forum: "CoT Vectors: Transferring and Probing the Reasoning Mechanisms of LLMs"
_ICLR.cc/2026/Conference — ICLR 2026 Poster_

### Official Review · Reviewer_QPGb · 2025-10-27

**Soundness:** 3
**Presentation:** 3
**Contribution:** 2
**Rating:** 6
**Confidence:** 3

**Summary:**

This paper presents CoT Vectors, a novel extension of task vectors for modeling multi-step reasoning in LLMs. The approach is well-motivated and provides meaningful insights into reasoning dynamics, with Learnable CoT Vectors showing stronger and more stable gains than extraction-based methods. However, the variability across intermediate layers suggests limitations in capturing intra-task diversity. Further exploration of adaptive or hierarchical vectorization could enhance robustness and generalization. Overall, this is a valuable contribution to understanding structured reasoning in LLMs.

**Strengths:**

The paper’s strengths lie in its novel approach to vectorizing Chain-of-Thought (CoT) reasoning, offering a compact and efficient representation of multi-step reasoning processes. The introduction of a teacher–student framework for learning CoT vectors is noteworthy, as it effectively captures the influence and transfer of reasoning knowledge. In addition, the paper is supported by comprehensive experimental evaluations, which clearly demonstrate the advantages and robustness of the proposed method across different settings.

**Weaknesses:**

Here are some concerns and/or issues about this work:

The proposed method is intriguing, but its systematic assessment of effectiveness is lacking. Specifically, it lacks a measurable way to categorize different types of CoTs and demonstrate when the proposed method works well and when it doesn’t.

The authors claim that the method saves compute cost because fine-tuning is not required. However, the inference process involves invoking both the frozen LLM and the student model. Given that models are typically used for inference far more often than training, the overall efficiency of the proposed method in real-world scenarios remains uncertain.

If I understand the authors correctly, there may be a hidden issue in Eq. 4. In Eq. 4, the extracted CoT vector takes the average. This means that if the answer is long and there are only a few mismatched tokens, the model may accept the result. However, if the problem is similar to the one depicted in Figure 1, a few mismatched tokens can invalidate the entire answer.

The experimental study lacks evaluation of the end-to-end efficiency of the proposed method. The authors propose comparing it to fine-tuning, so they should include some experimental results to support the significance of the proposed method compared to fine-tuning.

**Questions:**

See the weakness section.

---

> ### Author Response · Authors · 2025-11-20
> **Rebuttal for Reviewer QPGb**
>
> **W1: systematic assessment of different types of CoTs**
>
> We appreciate your insightful comment. Following the suggestion, we conduct a more systematic analysis to understand when our proposed CoT vector method works well and when it fails. Specifically, we used MMLU-Pro, which covers a broad range of subjects, as a representative multi-domain reasoning benchmark. For each subject category, we measure how applying the CoT vector changes accuracy, thereby revealing performance differences across distinct types of reasoning tasks. Results are shown in the table below:
> | Category | Baseline | Δ Accuracy (w/ CoT Vector) |
> |----------|----------|----------------------------|
> | biology | 60.71% | +5.81% |
> | history | 40.00% | +5.00% |
> | physics | 41.57% | +3.94% |
> | economics | 57.14% | +1.63% |
> | chemistry | 41.05% | +1.59% |
> | health | 51.95% | +0.65% |
> | other | 46.27% | +0.37% |
> | psychology | 62.67% | +0.33% |
> | engineering | 20.93% | 0.00% |
> | business | 52.31% | -0.38% |
> | math | 44.63% | -1.04% |
> | law | 32.58% | -1.40% |
> | philosophy | 27.03% | -4.06% |
> | computer science | 47.22% | -6.25% |
>
> Our analysis reveals a clear and consistent pattern across domains. Tasks involving structured, rule-based, and fact-driven reasoning—such as biology, history, physics, economics, and chemistry—show stable performance gains. These domains appear to share a coherent reasoning backbone that can be effectively captured by a single task-level vector.
>
> In contrast, performance decreases in domains requiring abstract reasoning, symbolic manipulation, or highly heterogeneous expertise, including computer science, philosophy, and law. These tasks lack a uniform reasoning structure, making it harder for a shared vector to generalize and sometimes causing the injected shift to introduce noise rather than useful guidance.
>
> **W2: overall efficiency of the proposed method**
>
> We thank the reviewer for raising this concern. We would like to clarify that our method does not require two models (frozen LLM and the student model) during training or inference. The “teacher” and “student” in our framework refer only to two different input sequences processed by the same frozen LLM, not two separate models. More clarification can be found in **"Q2" of General Response**.
>
> In terms of efficiency, the extracted vector requires no training at all, and its cost is limited to two forward passes of the frozen LLM.  The learnable vector has only a tiny number of parameters compared to LoRA (3.6K vs. 10M), making its optimization orders of magnitude cheaper than fine-tuning.
>
> At inference time, applying the CoT vector simply adds a lightweight shift to the hidden representations, which does not change the model size and adds virtually no computational cost, memory usage, or latency. More comparison of efficiency with LoRA can be refferd to our response to **W4**.
>
>
> **W3: average operation in Eq. 4 and concerns about mismatched tokens**
>
> Thank you for pointing out this potential issue. We would like to clarify that Eq. 4 does **not** average over the entire reasoning sequence. The averaging is computed only over the hidden states of **the final answer tokens**, and all intermediate CoT tokens are excluded.
> Moreover, both extraction and training use ground-truth (Q, CoT, A) triplets, ensuring that the answer tokens used for averaging always come from **correct reasoning traces**.
>
> Thus, the concern about “a few mismatched tokens being washed out by long sequences” does not arise, because the vector is derived solely from validated chain-of-thoughts.

---

> ### Author Response · Authors · 2025-11-20
> **Rebuttal for Reviewer QPGb**
>
> **W4: evaluation of the end-to-end efficiency**
>
> We thank the reviewer for this valuable suggestion. Our claim regarding efficiency is partially supported by Table 1 of the main text. As shown in Table 1, our Learnable CoT Vector consistently matches or surpasses the performance of the LoRA baseline while requiring three orders of magnitude fewer trainable parameters (e.g., 3.6K vs. 10.0M for Qwen).
>
> We have also added end-to-end efficiency evaluations here comparing our method with LoRA, the strongest PEFT baseline:
>
> | Method | Params | Training time | GPU| Epochs | Inference Latency  | Performance|
> |--------|--------|---------------------|----------|--------|----------------------|-----------------|
> | Learnable CoT Vector (Ours) | 3.6K | 3.97 min | 19.3 GB | 1-3 | 19.46 s | 83.5 |
> | LoRA | 10.0M (×2778) | 48.88 min (×12) | 19.69 GB (+2%) | 3-5 | 21.35 s (+9%) | 79.0 (-5.4%) |
>
> Training efficiency. Our Learnable CoT Vector uses three orders of magnitude fewer parameters than LoRA (e.g., 3.6K vs. 10M on Qwen). The new experiments further show that this parameter reduction leads to substantially faster convergence, lower training memory usage, and fewer required epochs, since only a single vector is optimized rather than large adapter modules.
> Inference efficiency. LoRA introduces additional matrix multiplications at every forward pass, whereas our method only performs a single vector addition at one layer. This introduces negligible computation, and our measurements confirm that inference latency is lower than LoRA-augmented models.
> These results demonstrate that our method is significantly more efficient than fine-tuning–based approaches in both training and inference.

---

> > ### Comment · Reviewer_QPGb · 2025-11-25
> >
> > I would like to acknowledge the authors’ rebuttal, particularly the detailed clarifications provided in response to the feedback. Given the positive assessment I previously offered, I will keep the overall score unchanged.

---

> > > ### Author Response · Authors · 2025-11-26
> > >
> > > We sincerely thank you for your valuable comments and constructive suggestions during the review. Your careful review and expertise have greatly improved our manuscript, and we are very grateful for the time and effort you dedicated to our work.

---

### Official Review · Reviewer_wcrC · 2025-10-28

**Soundness:** 3
**Presentation:** 3
**Contribution:** 2
**Rating:** 6
**Confidence:** 4

**Summary:**

This paper introduces CoT Vectors, a compact representation that encodes multi-step reasoning knowledge, enabling LLMs to enhance reasoning without modifying model weights.
Two approaches are proposed to obtain these vectors: extracted CoT vectors and learnable CoT vectors optimized via a teacher-student distillation framework, with analysis of their layer-wise effects.
Experiments show that the proposed methods can improve performance on the studied tasks.

**Strengths:**

1. The method proposed in the paper is overall clear in its conceptual approach and is presented in a way that is relatively easy to understand.
2. Layer-wise analysis offers a novel perspective on how reasoning in LLMs is internally organized.

**Weaknesses:**

1. The main experiments in the paper focus on mathematical reasoning and some domain-specific tasks, and the performance of the proposed method on more natural language logical reasoning tasks remains to be further validated.

2. The training of task vectors may depend on factors such as the number and quality of sampled support instances, as well as the balancing factor $\lambda$ between the two losses. Providing additional experimental results on these aspects could better demonstrate the robustness of the proposed method.

3. Although the authors provide some formula-based intuitive insights in Section 3.1, there remains a certain gap between the theoretical explanation and practical implementation. For example, the $\mu$ and shift vector in Equation 1 should ideally be functions that vary with the input, whereas the task-general CoT vectors seem to assume a single shared reasoning chain prompt (i.e., the same $K_C$ and $V_C$) for all instances of the same task. Even under this assumption, $\mu$ would still be expected to vary with the input.

4. While the proposed method introduces relatively few trainable parameters, in practice an additional teacher model is required to perform CoT reasoning and align the output distributions with the student model. Reporting only the number of trainable parameters may therefore provide an incomplete picture of the overall overhead.

5. The comparison in the paper is limited to LoRA as a parameter-efficient fine-tuning baseline, lacking evaluation against other fine-tuning or prompt optimization methods, such as Prefix-Tuning or Adapters.

**Questions:**

see Weakness.

---

> ### Author Response · Authors · 2025-11-20
> **Rebuttal for Reviewer wcrC**
>
> **W1: more natural language logical reasoning tasks**
>
> Thank you for the insightful suggestion. Following your comment, we have conducted additional experiments on two natural-language logical reasoning benchmarks: CommonsenseQA[1] and StrategyQA[2].
>
> CommonsenseQA focuses on short-horizon commonsense reasoning, while StrategyQA requires multi-hop, implicit natural language reasoning. The accuracy results are shown in the table below:
> | Method     | CommonsenseQA | StrategyQA |
> |------------|---------------|------------|
> | Baseline   | 72.7          | 60.8       |
> | Extracted  | 73.2          | 64.3       |
> | Learnable  | 73.7          | 65.0       |
>
> Consistent with our findings in mathematical reasoning, both Extracted CoT Vectors and Learnable CoT Vectors improve performance over the baseline on these tasks, with the learnable variant providing the larger gains. These results demonstrate that CoT Vectors are not limited to math-focused or domain-specific reasoning, but also generalize effectively to natural language logical reasoning.
> We will include the full results and analysis in the revised version (Table 1).
>
> **W2: number and quality of sampled support instances and the balancing factor $\lambda$ between the two losses**
>
> Thank you for raising this important point. We would like to clarify that we have already conducted extensive robustness and ablation analyses regarding (1) the number of sampled support instances, (2) the quality of CoT supervision, and (3) the balancing factor $\lambda$ between the two training losses. Due to page limits, these results were reported in the supplementary material (**Appendix A.5.5–A.5.7**). We summarize the key findings below for clarity.
>
> **(1) Number of sampled support instances (Appendix A.5.7).** Increasing the number of support instances leads to progressively stronger performance, for example, learnable CoT vector improves from 78.2 with 100 samples to 83.5 with 3000 samples, indicating that larger sets help the learnable CoT Vector capture more task-general reasoning patterns. Importantly, even with very small support sets (e.g., 100 samples), our method still produces substantial improvements over the base model, demonstrating strong data efficiency.
>
> **(2) Quality of sampled support CoT instances (Appendix A.5.5).**
> We compare extracted CoT vectors with human-labeled CoT (typically concise and logically clean) versus model-generated CoT (often longer and partially redundant). The performance difference is minimal (Figure 6), and both consistently yield gains. This shows that our method is robust to noisy or stylistically variable CoT supervision and does not rely on carefully curated chains.
>
> **(3) Balancing factor $\lambda$ between alignment loss and cross-entropy loss (Appendix A.5.6).**
> We study both simplified variants—using only the alignment loss or only the cross-entropy loss.
> Each variant still outperforms the baseline, indicating low sensitivity to $\lambda$. However, both underperform the full hybrid objective, confirming that the two losses play complementary roles. We also sweep $\lambda$  (0, 0.5, 1.0, and the only-CE setting), and $\lambda$=0.5 yields the best results, which is the value used in the main paper.
> | λ                     | Performance (%) |
> |-----------------------|-----------------|
> | 0 (only alignment loss) | 82.9            |
> | 0.5                   | 83.5            |
> | 1.0                   | 80.1            |
> | Only CE loss          | 78.4            |

---

> ### Author Response · Authors · 2025-11-20
> **Rebuttal for Reviewer wcrC**
>
> **W3:  $\mu$ and shift vector in Equation 1 should vary with the input**
>
> We thank the reviewer for this profound question, which touches upon the core hypothesis of our work. We agree that, according to the theoretical derivation in Equation 1, both the scaling factor \mu and the shift vector should ideally be sample-specific.
>
> $$
> \text{SA}(a, [K\_Q, K\_C, K\_A], [V\_Q, V\_C, V\_A]) = \text{SA}(a, [K\_Q, K\_A], [V\_Q, V\_A]) + \mu \cdot {\vec{v}\_{\text{CoT}}}
> $$
>
> The transition from this theoretical formulation to our practical implementation rests on a fundamental assumption, i.e., for a given task, there exists a shared, underlying reasoning pattern that is dominant and can be distilled into a task-general representation.
>
> While the instance-specific CoT vectors for each sample in the support set do vary, we hypothesize that their variations are not random noise but rather perturbations around a central, task-intrinsic "reasoning theme." By averaging (for extracted vectors) or by end-to-end optimization (for learnable vectors), we intentionally recover this shared component while smoothing out instance-level noise. Thus, the resulting vector does not encode one fixed chain of thought, but a compact approximation of the task’s most consistent reasoning strategy.
>
> Regarding the scaling factor $\mu$, we use a fixed value for extracted vectors for simplicity and stability. In the learnable setting, $\mu$ is implicitly absorbed into the learned vector itself—the optimization naturally adjusts both direction and magnitude to yield the most effective single shift, which negates the need for a dynamically changing $\mu$.
>
> Finally, the validity of using a task-general vector is supported by empirical evidence: across all evaluated benchmarks, a single learned vector per task reliably produces strong and consistent improvements. This suggests that the shared component of task reasoning is sufficiently dominant for a fixed vector to serve as an effective approximation of the theoretically instance-specific formulation.
>
> **W4: additional teacher model and training overhead**
>
> We thank the reviewer for raising this important point regarding computational overhead. We would like to clarify a potential misunderstanding: our framework does not employ two separate models. The "teacher" and "student" refer to two different forward passes of the same, frozen pre-trained LLM, processing different input sequences (with and without the CoT prompt). More clarification can be found in **"Q2" of General Response**.
>
> Specifically, for the Learnable CoT Vector, the teacher-student alignment is performed by comparing the hidden states from these two forward passes of the single frozen model, while only the tiny CoT Vector (e.g., 3.6K parameters) is optimized. This is fundamentally different from and vastly more efficient than fine-tuning methods like LoRA, which update millions of parameters (e.g., 10.0M).
>
> This extreme parameter efficiency directly translates into extremely low end-to-end overhead. As supplemented in our response to **"W4" of R.QPGb**, our Learnable CoT Vector converges much faster, requires substantially less training memory and fewer epochs than LoRA, precisely because we optimize a single vector rather than a massive set of adapter parameters.
> Therefore, reporting the minimal trainable parameters accurately reflects the overall efficiency advantage of our method, which is comprehensively demonstrated across both training and inference stages.
>
> **W5: More parameter-efficient fine-tuning baselines**
>
> Thank you for the suggestion. Following your comment, we have additionally included two widely used parameter-efficient fine-tuning (PEFT) baselines—Prefix-Tuning and IA³. Please refer to **"Q1" of General Response for more details**.
>
> [1] Talmor, Alon, et al. "Commonsenseqa: A question answering challenge targeting commonsense knowledge." Proceedings of the 2019 Conference of the North American Chapter of the Association for Computational Linguistics: Human Language Technologies, Volume 1 (Long and Short Papers). 2019.
>
> [2] Geva, Mor, et al. "Did aristotle use a laptop? a question answering benchmark with implicit reasoning strategies." Transactions of the Association for Computational Linguistics 9 (2021): 346-361.

---

### Official Review · Reviewer_YJ7G · 2025-11-02

**Soundness:** 3
**Presentation:** 3
**Contribution:** 3
**Rating:** 6
**Confidence:** 3

**Summary:**

The paper introduces CoT vectors, an extension of the task vectors to encode and transfer multi-step reasoning in LLMs. The authors propose and study two initiation of the CoT vectors: extracted CoT vectors and learnable CoT vectors. The extract CoT vectors are computed as the activation difference between reasoning and non-reasoning traces at a given layer; the learnable CoT vectors are parametrized as a learnable shift added to the hidden state of a layer and optimized in a teacher-student framework to distill reasoning representations. Experiments on three benchmarks (GSM8K, MATH, and MMLU-Pro) across two model families (Qwen2.5 Math and Llama3.1 8B) show consistent improvement over zero-shot CoT and LoRA baselines using only 3-4K parameters. Furthermore, the authors use CoT vectors to uncover a three-stage reasoning process in LLM and provide new insights that contrast with prior task vector findings. The observation is supported by additional analyses on layer-wise performance and latent space visualization.

**Strengths:**

- The paper has clear question formulation and motivation, is well-structured, and easy to follow.
- By probing with CoT vectors, the authors discover a U-shaped layer-wise performance curve and provide a three-stage reasoning process interpretation, offering novel insights that are different from prior task vector research. Supporting analyses provide a valuable contribution to understanding model internals.
- The proposed learnable CoT vectors address limitations of extraction-based methods (layer-wise instability), and effectiveness is supported through experiments and visualizations
- Evaluations span across multiple models, datasets, and baselines. The results demonstrate consistent improvement, validating the effectiveness of the proposed method.
- The method is parameter-efficient and requires negligible inference overhead, avoiding prompt lengthening and full-finetuning, which aligns with practical needs in efficient LLM deployment.

**Weaknesses:**

- The scope of "task-general" vectors is not fully supported by experiments. The CoT vectors are obtained per dataset, and the paper doesn't perform cross-task transferability tests to back the "task-general reasoning" claim. For example, the authors should consider applying vectors learned on GSM8K on MATH and vice versa.
- The improvements on Llama are small, and MMLU-pro only uses 70 annotated questions. It is difficult to assess the reliability of small gains without variance across seeds and significance tests.
- LoRA is the only parameter-efficient baseline, whereas other activation intervention methods are not compared or mentioned, including [1], [2], [3].
- The paper reports the best injection results. However, in practice, a comprehensive layer search could limit applicability.

[1] Azizi, Seyedarmin, Erfan Baghaei Potraghloo, and Massoud Pedram. "Activation Steering for Chain-of-Thought"
[2] Zhang, Jason, and Scott W. Viteri. "Uncovering Latent Chain of Thought Vectors in Large Language Models."
[3] Tang, Xinyu, et al. "Unlocking General Long Chain-of-Thought Reasoning Capabilities of Large Language Models via Representation Engineering."

**Questions:**

- In equation 7, why is the intervention applied to the hidden state rather than directly to the attention output? How do the author explain the difference between formulation and practical use?

---

> ### Author Response · Authors · 2025-11-20
> **Rebuttal for Reviewer YJ7G**
>
> **W1: Cross-task Transferability**
>
> Thank you for the insightful question. We agree that demonstrating transferability is essential for investigating whether CoT Vectors acquired from one source can be effectively applied to another.
>
> To address this, we have already conducted cross-dataset and cross-model transfer experiments (reported in **Appendix A.5.8**), and we will incorporate the key results in the revised main paper (**Section 4.2.4**).
>
> **(1) Cross-dataset transfer.**
> CoT Vectors learned on GSM8K improve performance on MATH (47.9 → 48.6), indicating that the representation captures general mathematical reasoning rather than dataset-specific features. We also observe consistent gains when transferring across domains , such as gains when transferring from MMLU-Pro to MATH (47.9 → 48.5).
>
> **(2) Cross-model transfer.**
> Vectors extracted using Qwen2.5-Math-7B-Instruct also improve Qwen2.5-Math-7B (74.6 → 77.5), showing that the learned direction is not tied to a specific model variant.
> These results directly support our claim: CoT Vectors demonstrate a degree of portability and generalization, remaining effective beyond the dataset or model used for their construction.
>
> **W2: Concerns about improvements on Llama and MMLU-pro**
>
> Thank you for raising this concern. We test on LLaMA (MMLU-Pro) with multiple random seeds and results are shown in the table below:
> | Seed | Baseline | Extracted | Learnable |
> |------|----------|-----------|-----------|
> | 42   | 44.6     | 45.5      | 46.2      |
> | 3407 | 43.2     | 44.2      | 44.9      |
> | 215  | 42.3     | 43.5      | 44.2      |
> | Mean±std | 43.4 ± 1.2 | 44.4 ± 1.0 | 45.1 ± 1.0 |
>
> Across all seeds, both Extracted and Learnable CoT Vectors consistently outperform the baseline, confirming that the gains are robust.
> The smaller improvement on LLaMA is explained in **Section 4.2.3** in main paper. Our layer-wise and PCA analyses show that LLaMA exhibits higher information density, weaker separation of reasoning stages, and lower variance explained by principal components, suggesting more entangled latent representations. These properties inherently limit how much a single direction can influence reasoning, so CoT Vectors still help but with naturally smaller margins.
> We also clarify that although only 70 MMLU-Pro instances contain ground-truth CoT for vector extraction, evaluation is performed on the full 1,000-question test set. The fact that the vector still yields consistent improvements from such a small support set highlights the efficiency of our method (Appendix A.5.7), consistent with prior work[1] showing similar sample sizes are sufficient for steering-vector extraction.
>
>
> **W3：Comparison with more PEFT baselines and activation intervention methods**
>
> Thank you for raising this concern. Following your suggestion, we have expanded our comparison beyond LoRA to include both PEFT baselines and activation-intervention methods.
>
> **(1) PEFT baselines.**
>
> We newly evaluate Prefix-Tuning and IA³, two representative PEFT methods belonging to distinct families. Please refer to **"Q1" of General Response** for more details.
>
> **(2) Activation-intervention methods.**
>
> We also compare our method against the activation-intervention approaches suggested in [1], [2], and [3].  Results are shown in the table below (reproduced where some code was unavailable):
> | Method | Performance(%) | Output Tokens |
> |--------|---------------|---------------|
> | Baseline | 74.6 | 1148 |
> | ASC[1] | 71.5 | 700 |
> | Zhang, et al.[2] | 74.0 | 1040 |
> | GLoRE[3] | 76.2 | 1217 |
> | Extracted CoT Vectors | 78.2 | 1105 |
> | Learnable CoT Vectors | 83.5 | 1046 |
>
> Method [2] is most closely related to our Extracted CoT Vector. As we have discussed in our Related Work section (line 145-152), it presents an early exploration of CoT-steering but relies only on basic extraction and offers limited analysis. Our work extends this direction by introducing a learnable CoT Vector and conducting comprehensive evaluations—including layer-wise studies, latent-space geometry, robustness, and transferability.
>
> Methods [1] and [3] target different objectives: [1] focuses on compressing CoT chains, while [3] aims to stimulate longer reasoning trajectories, both focusing on controlling CoT generation. In contrast, our method seeks to capture a task-general reasoning pattern shared across instances. Accordingly, our vector improves reasoning quality without significantly affecting output length, highlighting a conceptual distinction from these steer-vector approaches.
> We will include the discussion of [1] and [3] in the revised manuscript (Related works, line 145-152).

---

> ### Author Response · Authors · 2025-11-20
> **Rebuttal for Reviewer YJ7G**
>
> **W4: applicability and layer search**
>
> Thank you for highlighting this important and practical concern. We agree that a method requiring an exhaustive layer search would limit real-world applicability. In fact, this challenge is precisely one of the motivations behind our work. As discussed in the **main text (Lines 435–440)**, Extracted CoT Vectors indeed exhibit substantial layer-wise instability, with the optimal injection layer varying across tasks and models, making comprehensive layer search both costly and unreliable without supervision.
>
> This limitation is exactly why we propose Learnable CoT Vectors. Unlike extracted vectors, the learnable variant exhibits strong layer-wise stability. It consistently improves performance across almost all layers, with its strongest effect emerging in the shallowest layers—often simply the first layer. This property removes the need for any layer sweep in practice: simply injecting the vector into the first layer already achieves near-optimal results.
>
> Thus, rather than being restricted by layer-selection difficulties, our learnable approach explicitly resolves the issue.
>
> **Q1: attention output intervention**
>
> Thank you for the insightful question. Theoretically, as shown in Equation 7, the CoT Vector is most precisely defined at the attention-output level.
> In fact, we have also considered this issue. As discussed in **Appendix A.5.2** (Attention-level CoT Vector), we have implemented and evaluated the attention-output intervention suggested by the theoretical formulation. We also provide experimental results here:
> | Method     | GSM8K | MATH-E | MATH-H | MMLU-P | Avg.  |
> |------------|-------|--------|--------|--------|-------|
> | Baseline   | 74.6  | 69.9   | 47.9   | 33.2   | 56.4  |
> | Extracted  | 76.8  | 71.1   | 49.7   | 35.2   | 58.2  |
> | Learnable  | 76.1  | 71.9   | 48.6   | 34.1   | 57.9  |
>
> While it also yields performance improvements, its gains are consistently smaller than those achieved by activation-level (hidden states) injection. Therefore, the main paper presents the activation-level version, which gives the strongest results.
> We hypothesize that intervening directly on attention outputs introduces higher sensitivity and instability. This is likely because attention-level updates affect the fine-grained distribution over token-to-token interactions, making the learning signal more fragile. In contrast, activation-level intervention acts on the aggregated hidden representation and provides a more stable target for both extraction and optimization.
>
> Thus, although the formulation points to attention-level intervention, the activation-level implementation proves more stable and effective in practice. We will clarify this distinction more explicitly in the revision.
>
> [1] Azizi, Seyedarmin, Erfan Baghaei Potraghloo, and Massoud Pedram. "Activation Steering for Chain-of-Thought Compression."
>
> [2] Zhang, Jason, and Scott W. Viteri. "Uncovering Latent Chain of Thought Vectors in Large Language Models."
>
> [3] Tang, Xinyu, et al. "Unlocking General Long Chain-of-Thought Reasoning Capabilities of Large Language Models via Representation Engineering."

---

### Author Response · Authors · 2025-11-20
**General Response**

We gratefully thank all the reviewers for their valuable and constructive feedback. We are encouraged to see that the reviewers found our approach and analysis novel and insightful (Reviewer YJ7G, wcrC, and QPGb), appreciated the clarity of motivation and presentation (Reviewer YJ7G and wcrC), recognized the comprehensive experimental evaluation spanning multiple models and datasets (Reviewer YJ7G and QPGb), and acknowledged the practical efficiency of our method, achieving parameter-efficiency and negligible inference overhead without prompt lengthening or full-finetuning (Reviewer YJ7G).

Here, we have summarized the common questions and our responses as follows:

**Q1: Comparison with more PEFT baselines (R. YJ7G and R. wcrC)**

In addition to LoRA, we newly evaluate Prefix-Tuning[1] and IA³[2], two representative PEFT methods belonging to distinct families. Results are shown in the table below (Qwen-GSM8K):

| Method | #Params | Performance (%) |
|--------|---------|-----------------|
| LoRA | 10.0M | 79.0 |
| Prefix-Tuning | 573K | 75.0 |
| IA³ | 645K | 76.2 |
| Extracted (Ours) | 0 | 78.2 |
| Learnable (Ours) | 3.6K | 83.5 |

Across all evaluated reasoning benchmarks, our CoT Vector approach achieves superior or at least comparable performance relative to both Prefix-Tuning and IA³. Morever, Prefix-Tuning and IA³ introduce substantially larger tunable modules, whereas our method injects only a single compact vector, requiring orders of magnitude fewer trainable parameters. This confirms that CoT Vectors provide competitive improvements in reasoning ability while maintaining much stronger parameter efficiency.

**Q2: Misunderstanding about requiring two models for CoT Vector (R. wcrC and R. QPGb)**

We would like to clarify a potential misunderstanding: our framework does not use two separate models—such as a frozen teacher LLM and a student model—during either training or inference.
Instead, the terms teacher and student simply denote **two forward passes of the same frozen pre-trained LLM**, each receiving a different input sequence (with and without the CoT prompt).
To obtain the Extracted CoT Vector, we perform two passes of the frozen LLM using inputs with and without CoT, and compute the difference between the corresponding hidden states.
To obtain the Learnable CoT Vector, we insert a small trainable vector into a particular hidden layer and optimize it, such that the resulting student hidden state aligns with the teacher hidden state—again relying solely on a single frozen LLM.
During inference, the only required operation is to add the obtained CoT vector into the forward pass via simple vector addition.
Therefore, our method does not introduce any additional model beyond the original frozen LLM.

We also address other specific concerns in separate responses. We have uploaded the revised paper and highlighted the modified parts in blue.


[1] Li, Xiang Lisa, and Percy Liang. "Prefix-tuning: Optimizing continuous prompts for generation." arXiv preprint arXiv:2101.00190 (2021).

[2] Liu, Haokun, et al. "Few-shot parameter-efficient fine-tuning is better and cheaper than in-context learning." Advances in Neural Information Processing Systems 35 (2022): 1950-1965.

---

### Author Response · Authors · 2025-11-30
**Summary of Discussion Points for the Area Chair**

Dear Area Chair,

We sincerely appreciate your time and effort in handling our submission. We also thank all three reviewers for their constructive feedback and their unanimous initial assessment that the work is above the acceptance threshold.

Here is the summary of discussion points during the rebuttal period:

- Across the initial reviews, all reviewers recognized the novelty and empirical strength of CoT Vectors. Their key concerns focused on: cross-task transferability and robustness of improvements (`YJ7G`); comparison with more PEFT and activation-steering baselines (`YJ7G` and `wcrC`); evaluation on more natural language logical reasoning tasks (`wcrC`); theoretical grounding of the task-general vector assumption (`wcrC`); computational overhead and potential misunderstanding of the teacher-student framework (`wcrC` and `QPGb`); and systematic evaluation of effectiveness across different CoT types and end-to-end efficiency (`QPGb`).

- In our rebuttal, we provided comprehensive responses and new experimental results to address these concerns. (1) We demonstrated the cross-dataset and cross-model transferability of CoT Vectors and reported multi-seed results on Llama to validate the robustness of improvements. (2) We expanded comparisons to include Prefix-Tuning, IA³, and several activation-steering methods, highlighting superior parameter efficiency and key methodological differences. (3) We also justified the theoretical assumption with empirical evidence and clarified that the Learnable CoT Vector requires no layer search and uses only a single frozen LLM, thereby ensuring minimal overhead. (4) We also performed additional experiments on natural-language reasoning tasks (CommonsenseQA, StrategyQA), along with a systematic evaluation of effectiveness across different CoT types and detailed efficiency comparisons against LoRA.

- During the discussion period, Reviewer `QPGb` acknowledged our rebuttal and maintained a positive assessment, confirming that the concerns were addressed. For the other two reviewers, we have also provided thorough and evidence-backed responses to all raised points, though we have not yet received further feedback from them.

We will incorporate all clarifications, additional analyses, and updated results into the final version of the paper. With these improvements, CoT Vectors are further established as a parameter-efficient and broadly effective method for enhancing and probing the reasoning mechanisms of LLMs.

Thank you again for taking the time to read our summary.

Best regards,

Authors of Submission 9113

---

### Meta-Review · Area_Chair_Q5De · 2026-01-07

**Summary:**

The major concerns are as follows.

1) The  "task-general" vectors is not fully supported by experimental results. The generalization test is needed.
2) The improvements on Llama are small. And more experimental comparisons are needed to make the conclusion more convincing.
3) The overall efficiency of the proposed method in real-world scenarios remains uncertain.

**Reviewer Concerns:**

I think that most concerns have been replied by the authors.

**Reviewer Scores:**

I think the final scores should be 6 6 6.

---

### Decision · Program_Chairs · 2026-01-26

Accept (Poster)